



# Sea ice as a source of sea salt aerosol to Greenland ice cores: a model-based study

Rachael H. Rhodes [1], Xin Yang [2], Eric W. Wolff [1], Joseph R. McConnell [3], Markus M. Frey [2]

[1]Department of Earth Sciences, University of Cambridge, Cambridge, CB2 3EQ, UK
[2]British Antarctic Survey, Natural Environment Research Council, Cambridge, CB3 0ET, UK
[3]Division of Hydrologic Sciences, Desert Research Institute, Reno NV, 89512, USA

*Correspondence to:* Rachael H. Rhodes (rhr34@cam.ac.uk)

**Abstract.** Growing evidence suggests that the sea ice surface is an important source of sea salt aerosol and this
has significant implications for polar climate and atmospheric chemistry. It also offers the opportunity to use ice
core sea salt records as proxies for past sea ice extent. To explore this possibility in the Arctic region, we use a
chemical transport model to track the emission, transport and deposition of sea salt from both the open ocean and
the sea ice, allowing us to assess the relative importance of each. Our results confirm the importance of sea ice
sea salt (SISS) to the winter Arctic aerosol burden. For the first time, we explicitly simulate the sea salt
concentrations of Greenland snow and find they match high resolution Greenland ice core records to within a
factor of two. Our simulations suggest that SISS contributes to the winter maxima in sea salt characteristic of ice
cores across Greenland. A north-south gradient in the contribution of SISS relative to open ocean sea salt
(OOSS) exists across Greenland, with 50% of sea salt being SISS at northern sites such as NEEM, while only
10% of sea salt is SISS at southern locations such as ACT10C. Our model shows some skill at reproducing the
inter-annual variability in sea salt concentrations for 1991-1999 AD, particularly at Summit where up to 62% of
the variability is explained. Future work will involve constraining what is driving this inter-annual variability
and operating the model under different paleoclimatic conditions.

## 1 Introduction

Salty blowing snow lofted from the surface of sea ice may be an important source of sea salt aerosol to the polar
atmosphere (Yang et al., 2008), with significant implications for climate and atmospheric chemistry. Sea salt
aerosol act as cloud-condensation nuclei (O'Dowd et al., 1997) and ice nucleating particles (DeMott et al., 2016),
impacting radiative forcing (Murphy et al., 1998), as well as providing surfaces for heterogeneous chemical
reactions that impact the levels of key atmospheric trace gases, such as ozone (Knipping and Dabdub, 2003;





Yang et al., 2010). For paleoclimatogists, this new source of sea salt provides a mechanism that links the sea salt concentrations recorded in ice cores to sea ice extent, potentially validating the use of sea salt as a sea ice proxy (Abram et al., 2013).

Although early interpretations of ice core records assumed that sea salt was only sourced from bubble bursting at the ocean surface (e.g., Petit et al., 1999), two simple observations presented a paradoxical view: 1) seasonal sea salt maxima in most ice cores occur in winter not summer, 2) and sea salt concentrations are highest in glacial periods not interglacial periods. Given that sea ice extent is larger in winter relative to summer, and in glacials relative to interglacials, we would expect to see sea salt *lower* in winter and glacials if the open ocean was the only source of sea salt, due to the longer transport time between the open ocean and the ice sheet. Clearly that is not the case, and so, barring an unrealistic change in meteorological conditions, another source of sea salt must exist in winter (Wagenbach et al., 1998). Further evidence for an additional source comes from Antarctic snow chemistry that reveals reduced $Na^+:SO_4^{2-}$ values, relative to sea water, during winter months at Antarctic locations (Jourdain et al., 2008; Wagenbach et al., 1998). Unlike NaCl, which contains reactive $Cl^-$ (Keene et al., 1990; Röthlisberger et al., 2003), $Na_2SO_4$ is not fractionated in the atmosphere or following deposition, confirming that a *source* of fractionated sea salt exists in winter.

Sea ice fits the bill—its areal extent is greatest in winter, and its surface is covered by salty snow and frost flowers, which contain reduced $Na^+:SO_4^{2-}$ sea salt (Domine et al., 2004; Yang et al., 2008). Fractionation of $Na^+:SO_4^{2-}$ relative to sea water results from the precipitation of mirabilite salt ($Na_2SO_4 \cdot 10H_2O$) from brine in the channels that dissect the sea ice (Butler and Kennedy, 2015), and from sea water that floods or inundates slabs of sea ice (Massom et al., 2001). Frost flowers are now thought to make a relatively small contribution to the sea salt aerosol load sourced from the sea ice surface because of their high mechanical strength (Obbard et al., 2009), subsequent lack of observed aerosol production (Yang et al., 2017) under high wind speeds (Roscoe et al., 2011), and limited spatial and temporal range (Kaleschke et al., 2004; Perovich and Richter-Menge, 1994).

Yang et al.'s (2008) model proposes that principal source of sea salt from the sea ice surface is the entrainment of salty snow particles by high winds during blowing snow events, known to occur in the Antarctic (Mann et al., 2000; Nishimura and Nemoto, 2005) and Arctic (Savelyev et al., 2006). The air within the blowing snow layer is saturated, but the relative humidity reduces with height (Mann et al., 2000), allowing the water content of snow particles to sublime, generating sea salt aerosol (Déry and Yau, 2001). Empirical, field-based observations of sea salt aerosol generated by this phenomenon have only been obtained recently in the Weddell Sea (Antarctica) sea ice zone.





The sea ice source of sea salt aerosol appears to be critical for polar atmospheric chemistry. Domine et al. (2004) suggest that the salty snow on sea ice is an important source of Br$^-$ ions that contribute to the ozone depletion events observed over the sea ice in the spring. This idea is supported by evidence of air masses associated with ozone depletion originating from the sea ice zone (Jones et al., 2009). Yang et al. (2010) used a modelling approach to demonstrate that blowing snow provided the additional sea salt aerosol required to sustain the high levels of BrO responsible for the destruction of ozone in the polar regions.

To explore the implications of this additional source of sea salt aerosol for sea ice proxy development, a chemical transport model can be used to represent emission, transport and deposition of sea salt aerosol. Using this approach, Levine et al. (2014) found that sea-ice-sourced sea salt made a significant contribution to the winter sea salt aerosol budget at various Antarctic locations, and that this improved the model-data match with aerosol observations. Recently, these results have been replicated (Legrand et al., 2016) and confirmed using a different model (GEOS-Chem), but with similar parametrisations of sea salt emissions (Huang and Jaeglé, 2016). Huang and Jaeglé (2016) also argue for the importance of the blowing snow sea salt source in the Arctic region.

Here we investigate sea salt in the Arctic region in greater depth, with a particular emphasis on how sea-ice-sourced sea salt may impact the sea salt budget of Greenland ice cores. Doing so should help us to decipher whether Greenland ice core sea salt records have any potential to record past sea ice changes in the Arctic.

## 2 Methods

In this study, we use a chemical transport model to simulate sea salt aerosol emission, transport and deposition. We evaluate the model's performance by comparing simulations of monthly mean sea salt concentrations for 1991–1999 AD to values measured in the atmosphere and in Greenland ice cores.

### 2.1 Arctic sea salt aerosol data

We use sea salt aerosol data from five Arctic locations as targets for tuning our chemical transport model (Fig. 1). The five Arctic aerosol sites are Barrow in Alaska (Quinn et al., 2002), Alert in Canada (Barrie, 1995), Zeppelin Station on Svalbard, Villum Station in northern Greenland, and Summit on the Greenland ice sheet (see Supplement). For additional assessment of the model's skill at representing sea salt aerosol in the atmosphere, we also compare model output to measurements from five low-mid latitude aerosol sampling stations (Fig. S1) in the AEROCE-SEAREX network (Savoie et al., 2002). The age range of the aerosol data from each site is




displayed on any figure where the data are included. Where possible, we use data from the 1990s for comparison with model output. Aerosol data are compared to model output for 0.1–5 μm dry particle radius ($r_{dry}$).

## 2.2 Greenland ice core sea salt records

Ice core Na records from across Greenland are used to validate the performance of our chemical transport model

at simulating the concentration of sea salt deposited on the ice sheet (Fig. 1, Table 1). All the cores were analysed using the continuous melter system at the Desert Research Institute, Reno, USA (McConnell et al., 2002). Ice core data from 1990–1999 AD are compared with the model output encompassing the entire $r_{dry}$ range (0.1 to 10 μm).

For all of the ice cores, Na was measured by high-resolution inductively coupled plasma mass spectrometry (HR-

ICP-MS) with an estimated reproducibility of < 2 ppb (2 σ). The records are dated by annual layer counting of multiple chemical species that typically show different timings of seasonal maxima, e.g., sea salt, mineral dust and biomass burning products (Sigl et al., 2013). All the cores, except Tunu13, have accumulation rates > 200 kg $m^{-2}yr^{-1}$ (Table 1), providing monthly resolution records with age uncertainty of < 0.25 yr. Uncertainty on dating at the sub-annual scale originates from the uncertainty in the absolute timing of each seasonal marker and the

assumption of a constant annual snow accumulation rate. This dating method does not allow for seasonality in accumulation rate.

## 2.3 Chemical transport model

### 2.3.1 Model description

We use a simplified version of the Cambridge parallelised-Tropospheric Offline Model of Chemistry and

Transport (p-TOMCAT) to simulate the emission, transport and deposition of sea salt aerosol, following the work of Levine et al. (2014) (Fig. 2). p-TOMCAT is a 3D global model with a spatial resolution of 2.8° x 2.8° across 31 vertical sigma-pressure levels. Here we only describe changes to the model parameterisation implemented since Levine et al.'s (2014) study.

In this study, we drive p-TOMCAT with 6-hourly temperature, wind and humidity fields from the European

Centre for Medium-Range Weather Service Forecasts (ECMWF) ERA Interim reanalysis data set (Dee et al., 2011) whereas Levine et al. (2014) used ECMWF operational data. The significant precipitation bias of p-TOMCAT (Giannakopoulos et al., 2004), is remedied by applying a correction to force the simulated





precipitation values towards Global Precipitation Climatology Project (GPCP) observations (Adler et al., 2003), following Legrand et al. (2016).

Sea salt aerosol particles are traced from emission to deposition in 21 size bins ranging from 0.1 to 10 µm rdry. The ambient radius (rwet) of each particle may change each timestep according to relative humidity and temperature. Particles sourced from the open-ocean and the sea ice surface, which we will refer to as open-ocean sea salt (OOSS) and sea-ice sea salt (SISS) respectively, are treated separately, giving a total of 42 tracers. In p-TOMCAT, sea salt (SISS or OOSS) is assumed to be pure NaCl.

### 2.3.2 Sea salt emissions

Parameterisation of OOSS emissions follows Gong et al. (2003) and is based on the classic Monahan (1986) model of aerosol production via bubble bursting (Fig. 2). Gong's scheme is modified to account for a dependence of sea salt aerosol production on sea surface temperature (SST) (Eq. (4), Jaeglé et al. (2011)).

We use new salinity and particle size distributions for the blowing snow particles entrained from the sea ice surface (Fig. 2). Both reflect new observations made during a winter-time cruise of the RV Polarstern (June–August 2013) in the Weddell Sea, Antarctica, in the framework of the BLOWSEA project led by the British Antarctic Survey. The salinity distribution only includes measurements from the top 10 cm of the snow pack, as this snow is the most likely to be lofted up. The mean salinity is 0.30 psu, which is 14–fold lower than that of the salinity distribution used by Levine et al. (2014) (4.25 psu). The two-parameter gamma probability density function (Budd, 1966; Schmidt, 1982), which defines the size distribution of suspended particles in blowing snow events (Eq. (6), Yang at al. (2008)), has a snow particle radius of 70.3 µm and shape parameter (α) value of 2.

Yang at al.'s (2008) parameterisation of SISS production includes a snow age parameter (t in Yang at al.'s Eq. (5)). Levine et al. (2014) used a snow age of 5 days, but with our reduced snow salinity, this high snow age resulted in extremely low SISS emissions, so we adopted a value of 24 hr (see also Sect. 3.3.3).

Finally, the 'gustiness factor' of 1.18 used by Levine et al. (2014) to increase the 6-hourly wind speeds used for sea salt aerosol production has been removed because it is specific to a different chemical transport model (Gong et al., 2003). We haven't replaced this value so peak sea salt emissions may be underestimated due to the 6-hourly averaging of wind speeds.

### 2.3.3 Sea salt deposition

The deposition of OOSS and SISS in p-TOMCAT is based on the parameterisations of Reader and MacFarlane (2003) (see also Levine et al. (2014) Eq. (1-9). Wet deposition via nucleation and collision are both





parameterised by exponential decay (Eq. (1). Collision scavenging is determined by the collision scavenging parameter ($\alpha_C$, units: $m^2\ kg^{-1}$) that varies with $r_{wet}$ and by the rate of precipitation occurring at the same atmospheric level and all levels above ($PC_L$, units: $kg\ m^{-2}\ s^{-1}$). Nucleation scavenging is dependent on the nucleation scavenging parameter ($\alpha_N$, units of $m^2\ kg^{-1}$) and the rate of precipitation occurring only within the

same atmospheric level ($PN_L$, units: $kg\ m^{-2}\ s^{-1}$). Dry deposition only occurs in the surface layer of the model, which has a half-height (h, units: m) that varies between 23 and 36 m, depending on the geographic location and season. Calculation of the dry deposition velocity ($v_d$, Eq. (2), units: $m\ s^{-1}$) accounts for the processes of sedimentation and turbulence.

In order to compare our model simulations of Arctic sea salt aerosol to Greenland ice core Na concentrations, we

calculate how much OOSS and SISS is deposited at each time step, in addition to keeping track of the mass remaining in the atmosphere (M, units: kg). The mass of sea salt in each particle size bin ($r_{dry}$) removed from each sigma-pressure level (L) in the atmosphere at each time step ($\Delta t = 1800$ s) via wet (MWD, units: kg) and dry deposition (MDD, units: kg) is calculated by equations 1 and 2 respectively.

$$MWD_{L,rdry,t} = M_{L,rdry,t-\Delta t} \times e^{-(\alpha C\ PC_L + \alpha N\ PN_L)\ \Delta t} \tag{1}$$

$$MDD_{rdry,t} = M_{rdry,t-\Delta t} \times v_d \times \Delta t\ /\ h \tag{2}$$

$$SS_{mass,\ rdry,t} = MWD_{L,rdry,t} + MDD_{rdry,t} \tag{3}$$

$$Na_{mass,\ rdry,t} = SS_{mass,\ rdry,t} \times 0.3906 \tag{4}$$

$$Na_{flux} = (Na_{mass} \times 1e^9)\ /\ a \times 12 \tag{5}$$

$$[Na]_{snow} = Na_{flux}\ /A \tag{6}$$

After converting the mass of deposited sea salt ($SS_{mass}$, Eq. (3) to mass of Na (Eq. (4), the flux of Na ($Na_{flux}$, units: $\mu g\ m^{-2}\ yr^{-1}$) from the atmosphere to the ice sheet is calculated via Eq. (5), where a = area of grid box (units:

$m^2$) and $Na_{mass}$ is a monthly total Na mass deposited (units: kg). $Na_{flux}$ is then divided by the accumulation rate (A, units: $kg\ water\ m^{-2}\ yr^{-1}$), which is the sum of precipitation at all model levels, to give the simulated concentration of sea salt Na (either OOSS and SISS) in the snow (Eq. (6), $[Na]_{snow,}$ units: $\mu g\ kg^{-1}$ or parts per billion (ppb).



## 3 Tuning p-TOMCAT

### 3.1 Timing of sea salt deposition

Wet and dry deposition of sea salt in p-TOMCAT now takes place immediately after emissions, before any atmospheric mixing, which was not the case for Levine et al.'s (2014) study. This change was implemented to prevent large diameter aerosol, which can have atmospheric lifetimes (with respect to dry deposition) that are shorter than the model's dynamical time step (30 min), from leaving the surface layer. This modification caused only a modest difference in sea salt loading of the surface layer of the atmosphere in p-TOMCAT, particularly inland (Fig. S2A). However, the simulated ice core Na concentrations ([Na]) decreased substantially, sometimes by more than two thirds (Fig. S2B), because large aerosol were rapidly removed from the atmosphere after emission (Fig. S3), before they could be advected up above the surface layer.

### 3.2 Open ocean emissions

Comparison of the monthly aerosol sea salt data from the five mid-low latitude aerosol coastal sampling sites with p-TOMCAT simulations informs us about how well OOSS emissions are represented in the model. Overall, p-TOMCAT performs well, achieving normalised root mean squared differences (NRMSD) between 28 and 62 % at the five sites (Fig. S1). It is interesting that Chatham Island and Invercargill are only 1,300 km apart at similar latitude and yet aerosol [Na] is over-estimated for one site and under-estimated for another. Aerosol [Na] values tend to be under-estimated by p-TOMCAT, but usually the 1 σ inter-annual variability ranges of model and data overlap with each other. The tendency towards under-estimation could be due to: 1) OOSS emissions may be under-estimated due to 6-hourly averaging of wind speeds; and 2) depositing sea salt directly after emissions causes a strong depletion of large sea salt aerosol particles (> 4 μm $r_{dry}$) in the surface layer relative to the size spectrum of particles emitted (Fig. S3)—this deposition scheme may be too aggressive.

At the Arctic aerosol sampling sites, except Zeppelin, simulated OOSS Na concentrations fall within the range of observations in the summer months (Fig. 3). This suggests that p-TOMCAT captures OOSS in the Arctic well, assuming the model is accurate in simulating a minimal SISS contribution to the summer sea salt budget. For this reason, we have not tuned the OOSS parameterisation further.

### 3.3 Sea ice surface emissions

Simulated OOSS alone cannot reproduce the seasonal variability of aerosol Na observations at Arctic aerosol sites (Fig. 3). In the winter months, the simulated OOSS Na profiles show a deficit of Na relative to the

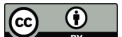



observations. This is consistent with the idea that blowing snow from the sea ice surface (SISS) is an important source of sea salt to the Arctic and its inclusion in model studies is essential to replicate Arctic aerosol observations.

We now consider the influence of some of the various parameters that can influence SISS emissions via the blowing snow mechanism. p-TOMCAT was run repeatedly for the year 1997 AD, changing individual parameters to assess the effect (Fig. 4).

### 3.3.1 Snow salinity

The snow salinity distribution used to generate SISS emissions in p-TOMCAT is based on measurements conducted over Antarctic sea ice (Sect. 2.3.2). To first order, we expect Arctic snow to be more saline than Antarctic snow, either because Arctic sea ice is generally thicker and suffers less flooding from sea water (Massom et al., 2001) or because less precipitation occurs in the Arctic sea ice zone relative to the Antarctic sea ice zone (Yang et al., 2010). Some estimates have put Arctic snow salinity as 3-fold higher than Antarctic snow salinity (Yang et al., 2010), whereas we use a 2-fold increase because the precipitation rate over sea ice simulated by p-TOMCAT is 50% higher over Antarctic sea ice relative to Arctic sea ice.

We tested the effect of using a higher snow salinity (3-fold Antarctic salinity) (Fig. 4D) and found that this produced a small reduction in the overall model-observations agreement across the five sites relative to the 2-fold Antarctic salinity case (Fig. 4C), although Summit, Barrow and Villum all showed a reduced model-data difference (ΔNa). There are very few measurements of snow salinity on sea ice in the Arctic to compare to, and values are likely to vary with season and location. However, Mundy et al. (2005) reported a mean salinity of 0.1 psu for the surface snow in the central Canadian Arctic. This value is close to the median of the Weddell Sea salinity distribution multiplied by two, 0.12 psu, so we do not adopt the 3-fold increase in snow salinity relative to Antarctica.

### 3.3.2 Multi-year sea ice

In the Arctic, around half of the winter sea ice is multi-year ice. We know that the salinity, or brine content, of sea ice decreases as brine is progressively expelled through brine rejection (Cox and Weeks, 1974). Therefore, it is likely that the brine supply to the sea ice surface reduces with time, thereby reducing the salinity of the surface snow. Furthermore, multi-year sea ice is generally thicker and more stable than first-year ice, which limits flooding and inundation by sea water at cracks and leads (Massom et al., 2001), also likely reducing snow salinity as the salt supply is replenished less often. We have little direct evidence but we can surmise that first-year sea





ice will harbour more saline snow than multi-year sea ice, therefore producing blowing snow particles with a higher Na concentration.

In p-TOMCAT, Arctic sea ice in each grid box is classed as multi-year ice if it was present in the preceding September. As we have little field evidence to indicate how snow salinity evolves with time, we crudely reduce the SISS emissions of regions covered by multi-year ice instead of explicitly altering the salinity of blowing snow particles above multi-year ice. We tested 3 different scenarios: 1) both first-year and multi-year sea ice contribute equally to SISS emissions, i.e., there is no distinction between first-year and multi-year sea ice (Fig. 4A), 2) only first-year sea ice contributes SISS emissions (Fig. 4B), and 3) multi-year sea ice contributes 50% of the SISS emissions of first-year sea ice (Fig. 4C). For 1997 AD, the total SISS emissions in the Arctic region in each experiment were 3.57, 1.67 and 2.66 Tg Na respectively. The impact on simulated Na concentrations at Arctic aerosol monitoring sites was significant at all five Arctic sites (Fig. 4 A-C). At Zeppelin, OOSS is over-estimated by the model so ΔNa is always positive regardless of the multi-year ice option. p-TOMCAT simulates too much Na (positive ΔNa) at Villum in N. Greenland when all sea ice contributes the same SISS emissions and not enough Na (negative ΔNa) when only first-year ice contributes. The ΔNa value is lowest when multi-year ice contributes 50% of SISS emissions.

We chose to adopt option (3) so that multi-year sea ice contributes 50% of the SISS emissions of first-year sea ice because it produces the lowest NRMSD across the Arctic sites. Although this option does not produce the best correspondence between model and observations at all the Arctic aerosol sites, it is important to make some distinction between the SISS emissions of first- and multi-year sea ice. However, we do note that the difference between the simulated seasonal aerosol [Na] at the five sites under the three different multi-year sea ice options is relatively small at Summit (Fig. S4) and therefore this choice does not greatly impact the sea salt budget of the atmosphere above Greenland.

### 3.3.3 Snow age

Higher values of snow age result in reduced SISS emissions. We tested the impact of decreasing the snow age from 24 hr to 12 hr in the Arctic. For some sites, such as Barrow and Villum, ΔNa was reduced with the lower snow age (Fig. 4E compared to Fig. 4C). The model-observations match across all the Arctic sites was reduced for the lower snow age (NRMSD increased), but if we exclude Zeppelin, the NRMSD is similar that achieved using a snow age of 24hr. The maximum change in monthly [Na] caused by halving the snow age is a 25% increase in [Na] at Barrow (Fig. S4).



Lack of observations of snow on sea ice in the Arctic, and of sea salt aerosol produced during blowing events, makes it difficult to constrain many of the key parameters related to the blowing snow SISS emission process. Although we have chosen a snow salinity distribution double that of Antarctic observations, a snow age of 24 hr and a 50% reduction in SISS emissions from multi-year sea ice relative to first-year sea ice, we understand that a
different combination of these parameters could effectively produce the same results.

### 3.4 Importance of sea-ice-sourced sea salt aerosol

Despite these somewhat ambiguous choices of tuning parameters, it is important to note that in all five sensitivity tests conducted for 1997 AD, SISS contributes to offset the winter OOSS Na deficit at all five Arctic aerosol sites (Fig. S4). For the full 1991–1999 AD simulations using the tuned parameters, the addition of SISS produces
seasonal cycles that match the Arctic aerosol observations well with RMSDs of between 34% for Villum and 90% for Alert (Fig. 3). At Zeppelin on Svalbard, the modelled OOSS contribution is too high throughout the year. However, the seasonal profile of SISS looks promising. Its amplitude is similar to the seasonal cycle of the observations. Villum, N. Greenland, shows the best model-observations agreement, with SISS contributing 83% of the total Na in the winter months on average. Results for Barrow, Alaska, are equally encouraging for January
to June, but p-TOMCAT appears to underestimate SISS in the latter half of the year.

Only Alert, Canada, shows a significant offset between the aerosol observations and the modelled Na concentration (Fig. 3). The summer concentrations, dominated by OOSS match well, but in other months p-TOMCAT underestimates [Na]. Huang and Jaeglé (2016) had a similar problem estimating aerosol [Na] at Alert and suggested that it results from Alert being situated in a region of relatively calm and stable meteorological
conditions where the threshold wind speed (~7 m s$^{-1}$) for SISS emissions is not reached as often. Huang and Jaeglé (2016) found that the inclusion of an explicitly parameterised frost flower source (Xu et al., 2013) helped to match the observed sea salt aerosol budget at Alert, but further field work is required to assess to what extent frost flowers contribute aerosol to the atmospheric sea salt budget at low wind speeds in reality.

The simulated seasonal Na aerosol cycle for Summit, Greenland, matches the aerosol observations well (Fig. 3).
Our results suggest that OOSS is the dominant source of Na to the high altitude central interior of the Greenland ice sheet with significant SISS Na only present from November to May, contributing a maximum of 33% of the monthly Na budget.





## 4 Comparison of p-TOMCAT simulations to ice core Na records

We now compare our p-TOMCAT simulations of deposited sea salt for 1991–1999 AD, produced using the tuning specified above, to investigate the contribution of SISS to sea salt concentrations of Greenland snow and ice core records. All the ice cores we consider are located at > 2000 m elevation and > 100 km inland (Table 1) so maximum Na concentrations are < 100 ppb. Seasonal variability in [Na] is consistently characterised by winter maxima and summer minima (Fig. 5); the amplitude of the mean seasonal cycle in the different ice cores varies between 6 and 55 ppb.

### 4.1 Influence of accumulation rate

Given that simulation of ice core Na concentrations using p-TOMCAT requires both the mass of Na deposited and the amount of precipitation at the ice core site (Eq. (6)), it is important that p-TOMCAT simulates precipitation accurately over the polar regions. On the annual scale, the p-TOMCAT precipitation rates (forced towards GPCP values, Sect. 2.3.1) agree well with ice core accumulation rates (Fig. 6A). Northern sites like NEEM and Tunu show model-ice core agreement to within 30%. Summit annual mean accumulation rate is estimated to within 2%. Further south, the model-ice core agreement reduces as p-TOMCAT has trouble capturing the steep gradient in accumulation rate between the coast and the interior of the ice sheet over Southern Greenland. At ACT11d, for example, the simulated accumulation rate is 250% higher than that suggested by the ice core.

The model-calculated precipitation at a single Greenland ice core site can vary by a factor of 4 across a year (Fig. 6B–D). At NEEM in northwest Greenland, model-calculated precipitation rate is consistently higher in summer relative to winter (Fig. 6B), whereas at Summit in central Greenland model-calculated precipitation rate is greater in winter relative to summer (Fig. 6C). Ice core sites further south don't show a clear seasonal signal in model-calculated precipitation rate (Fig. 6D). We have a small amount of information about how accumulation rates over Greenland vary seasonally. Recent field measurements at Summit (2003–2014 AD) agree with satellite-based laser altimetry measurements, indicating that the monthly accumulation rates are highly variable with no significant seasonal difference but a tendency towards relatively low accumulation in spring and high accumulation in autumn (Fig. 6C). Other work, focused on the Summit, NGRIP and NEEM sites, found evidence for a summer-weighted bias in accumulation (Shuman et al., 1995, 2001; Steen-Larsen et al., 2011), suggesting p-TOMCAT may in fact be doing a good job of representing seasonal accumulation variability in northern Greenland.





We should remember that a constant rate of accumulation per year was assumed when dating the ice core records. With this in mind, we test the effect of substituting the constant monthly ice core accumulation rate for A in Eq. (6) when calculating the Na concentration of snow falling at the ice core sites. This does not account for all possible bias due to precipitation seasonality in the model because wet deposition of sea salt is controlled by

precipitation occurrence and amount (Eq. (1)). Simulated ice core [Na] calculated by this method are displayed on Fig. 5B, and simulated ice core [Na] calculated using the model-calculated accumulation rate in Eq. (6) are displayed on Fig. 5A.

At ice core sites where accumulation rates are over-estimated by p-TOMCAT, i.e., D4, D5, Das2 and S. Greenland (ACT10C, ACT3 and Das1), Na concentrations broadly increase when the [lower] ice core

accumulation rate is used (Fig. 5B compared to Fig. 5A). Modelled precipitation for Summit (Fig. 6C), D4, D5 and Das2 is reduced in April to June relative to other months causing a prominent spring-early summer maximum in simulated Na, specifically OOSS (Fig. 5A). Using the constant ice core accumulation rate this feature disappears and the [Na] maximum occurs in the winter months, in agreement with the ice core data seasonality (Fig. 5B). Moreover, peak [Na] is then driven by the addition of SISS, rather than increased OOSS. It is possible

that the assumption of winter timing of [Na] peaks made in ice core dating is incorrect and that [Na] seasonality in Greenland ice cores is actually like the simulated profiles on Fig. 5A. However, this seems unlikely because [Na] values of Greenland aerosol (Fig. 3) and fresh surface snow at Summit (Fig. S5) peak in the winter months.

## 4.2 Smoothing of the snowpack Na signal

Comparison between p-TOMCAT [Na] simulations and Greenland ice core records reveal significant month-to-

month variability in the simulated time series that is not present in the ice core records, which are all characterised by smoothly oscillating [Na] with a clear seasonality (Fig. 7). We hypothesise that the deposited Na signal is smoothed by surface snow redistribution by winds and compaction of the snow pack during densification (Dibb and Jaffrezo, 1997). Evidence for this smoothing process comes from comparison of [Na] measurements of surface snow samples at Summit and ice core [Na] measurements dating from the same time

interval (Fig. S5). The surface snow [Na] is much more variable with rapid oscillations in concentrations, but the overall seasonal cycle corresponds well with the ice core record. The ice core [Na] signal may also be damped by dispersive mixing within the continuous analysis system (Breton et al., 2012), specifically for lower accumulation sites such as Tunu. We crudely represent the cumulative effect of these smoothing processes by applying a Savitzky-Golay filter (span = 4%, order = 2) to the simulated [Na] time series (Fig. 7). The stacked



simulated [Na] seasonal cycles for the years 1991–1999 AD are displayed on Fig. 5. Unfiltered Na seasonal cycle stacks are displayed on Fig. S6.

### 4.3 How well are Greenland ice core records represented by p-TOMCAT?

### 4.3.1 Annual mean

The majority of Greenland ice core annual mean [Na] values (1991–1999 AD) are simulated to within a factor of 2 by p-TOMCAT (Fig. 8A, Table 2). Tunu, NEEM and ACT10C annual means are simulated most accurately, regardless of the accumulation rate used to calculate the simulated [Na] (Table 2). Das2 in southeast Greenland and ACT11d and ACT2 is southwest Greenland are the most poorly simulated with p-TOMCAT over-estimating the extremely low ice core annual mean [Na] values of 5–8 ppb by as much as 750%. p-TOMCAT severely

over-estimates the accumulation rate for these sites (Fig. 6A), suggesting that too much sea salt is deposited by wet deposition.

### 4.3.2 Seasonal cycle

p-TOMCAT estimates for summer (JJA) [Na] are higher than the ice core data (Fig. 8C, Table 2), sometimes by a factor of 5 or more, but we note that summer ice core [Na] values can be as low as 1 ppb. It is interesting that the

summer OOSS contribution to the ice core budget is over-estimated by p-TOMCAT because simulated aerosol OOSS concentrations in the surface layer of the atmosphere appear to match summer observations well (Fig. 3). We suspect this results from the simplistic deposition scheme of p-TOMCAT, which allows super-micron sized OOSS particles to be transported to the ice sheet and wet-deposited from high levels in the atmosphere (Fig. S3). The deposition scheme does not include parameterisation of below-cloud scavenging (Zhang et al., 2013) and

differences between snow and rain wet deposition rates (Wang et al., 2014) found in other chemical transport models, or explicit consideration of fog deposition, which is common on the Greenland ice sheet (Bergin et al., 1995). Simulated winter (DJF) [Na] values calculated using the p-TOMCAT precipitation rate appear to match the ice core values well at all sites (Table 2), but this is deceptive because the simulated seasonal maxima occur later in the year (Fig. 5A). p-TOMCAT winter [Na] values calculated using the ice core accumulation rates are

only within a factor 2 of the ice core values for Summit, Tunu, NEEM (all cores) and ACT10C (Fig. 8B, Table 2).

Despite some problems in accurately simulating the absolute seasonal [Na] values, p-TOMCAT does a good job at simulating the amplitude of the seasonal [Na] cycle, both in absolute units of ppb and the relative amplitude,





normalised to the summer minima [Na] (Table 2). This observation gives us confidence that p-TOMCAT is simulating meaningful seasonal variability.

### 4.3.3 Inter-annual variability

To test whether or not p-TOMCAT shows any skill at reproducing the inter-annual variability in ice core [Na] we
regress the annual mean [Na], annual maximum [Na] and the inter-annual [Na] difference of the ice cores against the equivalent values simulated by p-TOMCAT, calculated using both options for accumulation rate (Table S1). In many cases, the sign of regression is negative, or close to zero, indicating that p-TOMCAT has no skill at all. However, in five cases we obtain significant ($p < 0.05$) positive correlations between ice core data and model simulation. These results indicate that p-TOMCAT captures 54% and 43% of the inter-annual variability in the
annual mean [Na] and annual maximum [Na] respectively at Summit, and 62% of the year-to-year change in annual mean [Na]. 29% and 58% of the inter-annual variability in annual maximum [Na] at Tunu and NEEM-2008-S3, respectively, also appear to be captured by p-TOMCAT.

These results are promising, given that 1991 to 1999 AD is a relatively short time series for comparison. Additionally, it is unlikely that a chemical transport model could explain a greater proportion of inter-annual
variability in ice core [Na] than achieved here. This is because ice core chemistry records are affected by several factors that impact the final record preserved, in addition to the meteorology and source conditions parameterised by p-TOMCAT. Factors such as snow redistribution and wind-generated features such as sastrugi can cause chemistry (Gfeller et al., 2014) and accumulation rate (Mosley-Thompson et al., 2001) records from proximal ice cores to differ; Dibb and Jaffrezo (1997) found annual mean [Na] of the snowpack at Greenland varied by up
30% between sites < 1 km apart. We can see that this is case by comparing the different NEEM ice core records or S. Greenland ice core records on Fig. 5 that show significant differences in [Na] despite being located in the same p-TOMCAT grid box.

### 5. Importance of sea-ice-sourced sea salt for Greenland ice core records

Greenland ice core simulations suggest that SISS makes an important contribution to the sea salt budget during
winter and the shoulder seasons. Summer ice core Na concentrations apparently reflect only OOSS Na levels (Fig. 5). This agrees with Arctic aerosol observations and simulations (Fig. 3). For Summit, Greenland, we can model the Na loading of the surface atmosphere and the Na concentration of the deposited snow. Both agree that summer minima reflect OOSS and that winter maxima are supplemented by SISS.





SISS accounts for between 10 and 50% of the winter sea salt budget of Greenland ice cores (Fig. 8D, Table 2). The SISS:OOSS ratio is marked by a north-south gradient across Greenland as more northerly sites, closer to sea ice, show elevated SISS relative to OOSS (Fig. 8D). SISS:OOSS also increases to the west of Greenland where the prevailing wind comes across the sea ice of Hudson Bay. It is likely that the SISS:OOSS ratio was greater in the past when temperatures were cooler and Arctic sea ice was expanded, for example during the Last Glacial Period. Running p-TOMCAT with prescribed sea ice and meteorology for paleo-conditions will allow us to test this.

## 6. Summary

This study supports Levine et al. (2014), Legrand et al. (2016) and Huang and Jaeglé (2016) who all argue for the importance of a winter source of sea salt aerosol from the sea ice surface to the aerosol budget of the polar regions. We demonstrate that winter SISS is required, in addition to OOSS, in order to reproduce the magnitude and seasonality of aerosol observations of sea salt at five Arctic locations across Alaska, Canada, Greenland and Svalbard.

For the first time, we use a chemical transport model to explicitly simulate the Na concentration of snow deposited on the Greenland ice sheet to within a factor of 2. Our simulations for 1991–1999 AD suggest that SISS contributes to the winter maxima observed in all the ice cores, but that in some cases, OOSS alone can produce winter maxima and summer minima in sea salt in ice cores. A north-south gradient in the contribution of SISS to the total winter ice core sea salt budget is simulated across Greenland, with 50% of sea salt being SISS at NEEM and only 10% at southern Greenland sites, such as ACT10C. This spatial pattern hints that comparison between cores from northern and southern Greenland could help to isolate any independent change in SISS relative to OOSS. p-TOMCAT shows some skill in simulating the inter-annual variability of [Na] in the ice core records from Tunu, NEEM and particularly Summit, where 62% of the inter-variability in annual mean [Na] is captured by the model (Table S1). Future work will use the model simulations to assess what factor(s) is driving the inter-annual variability.

Our chemical transport model simulations suggest that [Na] records from Greenland ice cores can only inform us about winter, or maximum seasonal sea ice extent, in the Late Holocene. In the summer months, SISS contributions to the sea salt budget are virtually zero so any change in summer sea ice extent over time is unlikely to be recorded, unless the ratio of SISS to OOSS is changed substantially. Other ice core proxies, such as



methanesulfonate, which is linked to primary productivity in the surface ocean, should be considered for reconstructing summer sea ice conditions (Maselli et al., 2017).

More Arctic observations of blowing snow events (particle size and chemical composition) and snow on the sea ice surface (salinity and its seasonally-resolved evolution with time) are required before process-based modelling of blowing snow SISS emissions can be improved.

**Supplement link**

Figures S1–S6, Table S1, and Greenland ice core Na data are located in the Supplement.

**Data availability**

All model simulations produced in this study are available as NetCDF files on request from RH Rhodes (rhr34@cam.ac.uk). Greenland ice core data are available at: https://arcticdata.io/ or in the Supplement. Arctic aerosol data used in this study from Alert (Canada), Zeppelin Station (Svalbard) and Villum Station in N. Greenland are available at: http://ebas.nilu.no/. Summit (central Greenland) aerosol and surface snow Na data are available at: https://arcticdata.io/catalog/_-_view/urn:uuid:e9136a64-661f-470d-9b3a-72f31d54d066. Aerosol chemistry data from the AEROCE-SEAREX networks are available at http://aerocom.met.no/download/AEROCE-SEAREX/.

**Acknowledgements**

This work was supported by a European Commission Horizon 2020 Marie Sklodowska-Curie Individual Fellowship (No. 658120, SEADOG) to RH Rhodes. EW Wolff is supported by a Royal Society Professorship. X Yang and M Frey gratefully acknowledge financial support from Natural Environment Research Council (UK) through the BLOWSEA project (NE/J023051/1). We thank W Feng and M Chipperfield at University of Leeds for support in using ERA-interim data and the ICT Unix team at the British Antarctic Survey for their help running p-TOMCAT. The Greenland ice cores were collected and analysed under numerous NSF Office of Polar Programs grants including 021515, 0856845, 0909499, 0909499, and 1204176, as well as NASA grants NAG512752 and NAG04GI66G. This international collaboration was supported partially by NSF grant 0968391. We are grateful to the students, staff, and numerous collaborators of the Desert Research Institute ultra-trace ice core chemistry laboratory for help in collecting and analysing the ice cores.



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




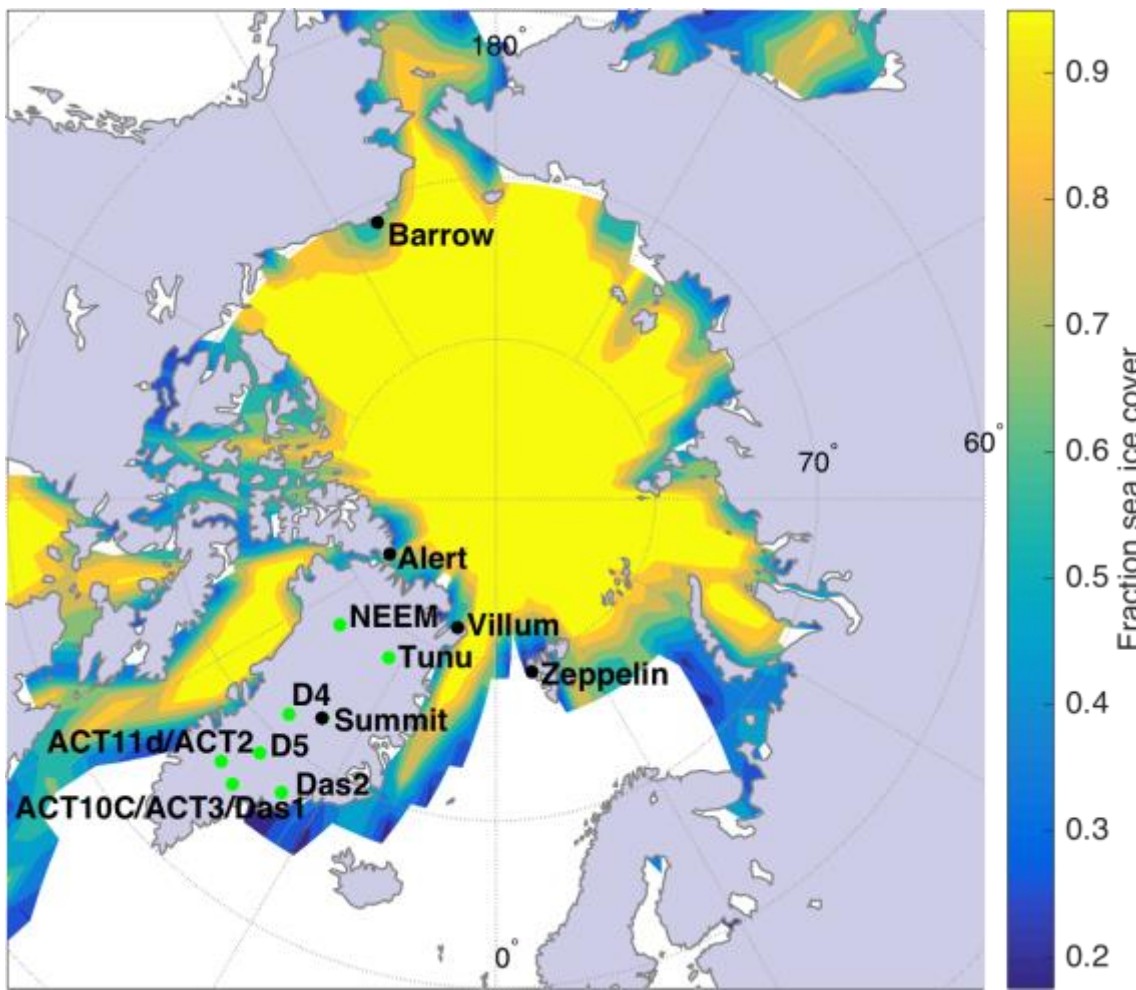

**Figure 1: Map of the Arctic region showing locations of aerosol sampling stations (black circles) and ice cores (green circles) used in this study. Contoured shading is mean February fractional sea ice coverage for 1991-1999 AD, as prescribed in p-TOMCAT.**




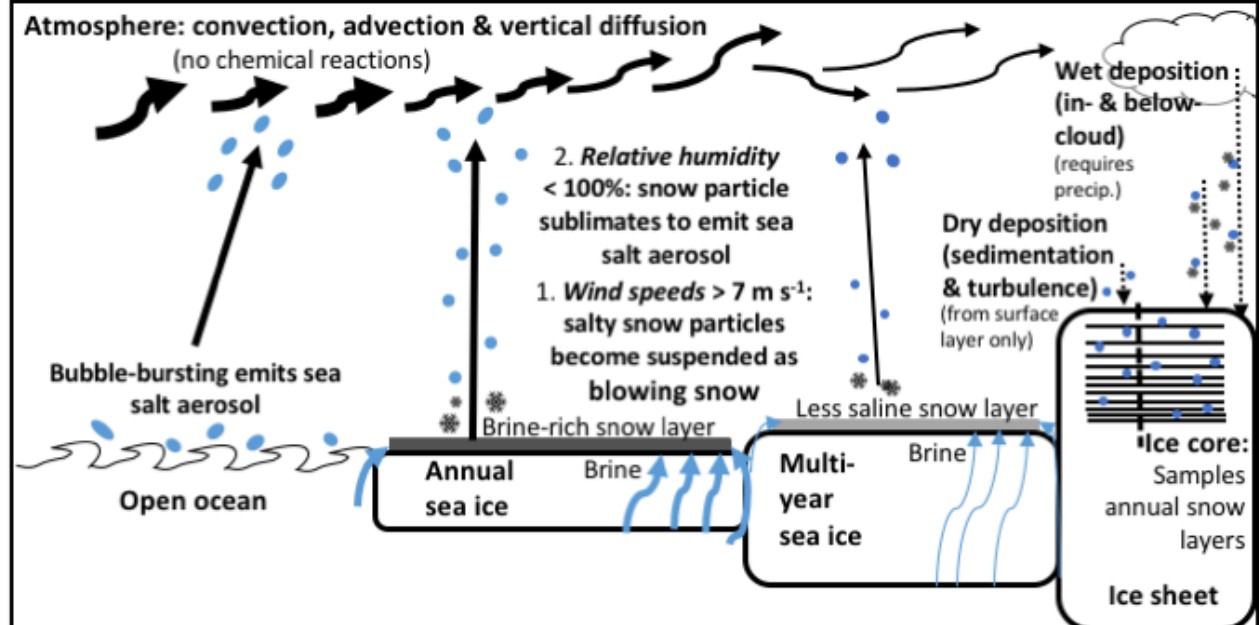

**Figure 2: Schematic of processes parameterized by p-TOMCAT that influence sea salt concentrations in the atmosphere and ice cores.**



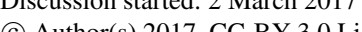



**Figure 3: Sea salt Na aerosol concentrations at Arctic locations simulated by p-TOMCAT for 1991–1999 AD compared to observations. Observations and model results are mean monthly values with uncertainty bars or shaded bounds representing ± 1 σ of the inter-annual variability. The Summit observations are not plotted with uncertainty bars because temporal coverage of data set is too poor (Supplement). p-TOMCAT aerosol size bins 1–18 (0.1–5 μm $r_{dry}$) are included. Dates of observations are displayed on each subplot.**





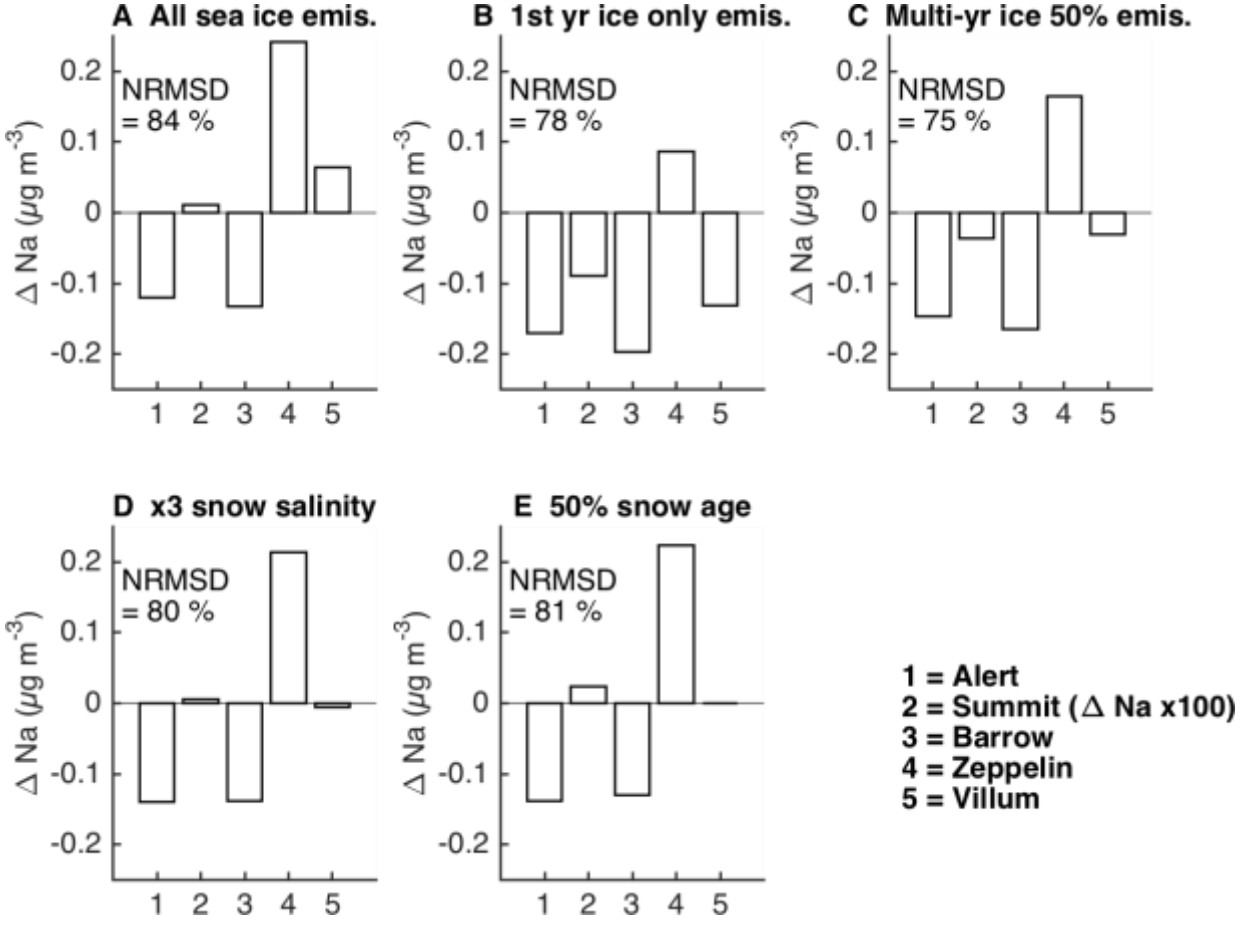

**Figure 4: Sensitivity of p-TOMCAT Na aerosol simulations at 5 Arctic locations to parameters associated with SISS emissions via blowing snow. Each panel (A-E) displays the mean difference between monthly (not including July-September) model results and observations (ΔNa) for each site. Positive [negative] values indicate that p-TOMCAT over- [under-] estimates aerosol Na concentration. The normalised root mean square difference (NRMSD) between model and data is calculated for each of the 5 sites and the mean NRMSD across all 5 sites is displayed on each panel.**




**Figure 5: Mean monthly mean sea salt Na concentrations of Greenland ice cores simulated by p-TOMCAT for 1991–1999 AD compared to data.**
OOSS simulations (red dashed line) and combined OOSS and SISS simulations (blue line) are shown with uncertainty envelopes (red and blue shading respectively), representing ±
1 σ of the simulated inter-annual variability. Mean monthly ice core Na concentrations (green) are shown with uncertainty bars denoting ± 1 σ of the inter-annual variability. For
two ice core locations, three different ice core records are plotted, as indicated by the legend. Two different options for simulated sea salt concentrations are displayed: A)
[Na] calculated using p-TOMCAT precipitation output in Eq. (5), B) [Na] calculated using the constant annual accumulation rate indicated by the ice core records (Table 1) in
Eq. (5). In both cases, the p-TOMCAT monthly mean time series has been smoothed using a Savitzky-Golay filter (span 4%, order 2) prior to stacking of the monthly mean
values for 1991–1999 AD (Sect. 4.2).



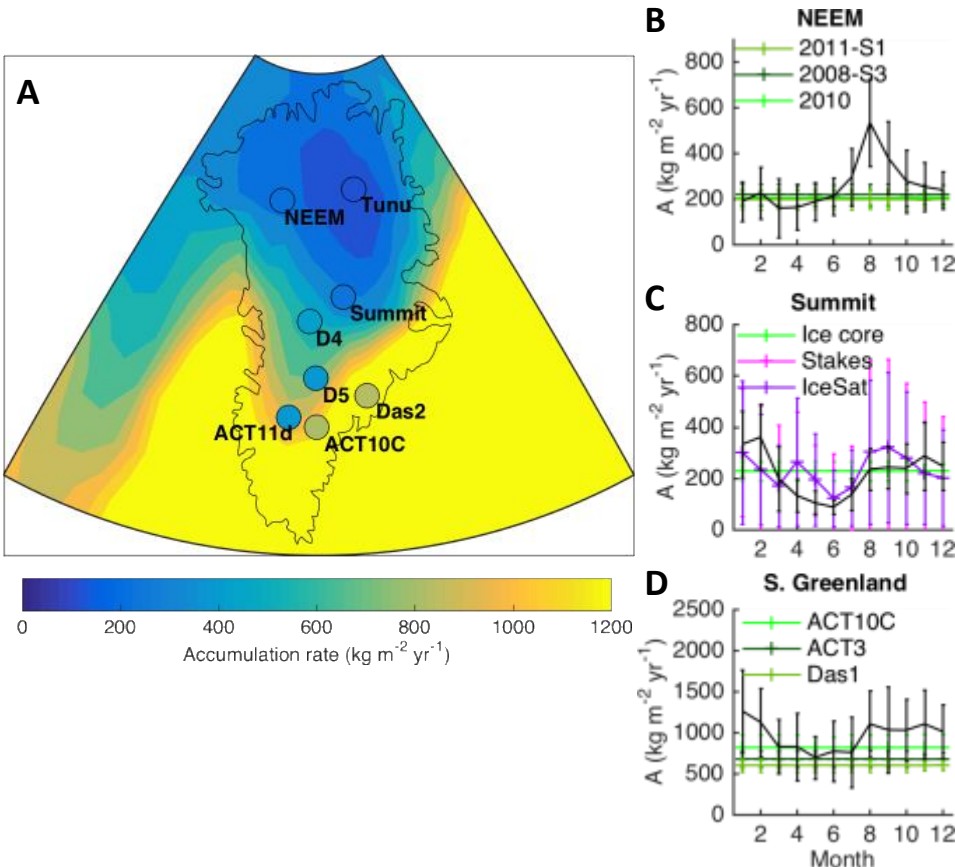

**Figure 6: Comparison between Greenland ice core and snow accumulation rates and simulated precipitation for 1991-1999 AD. A)** Map of Greenland showing contoured simulated annual accumulation rate. Actual ice core accumulation rates for 1990s are shown as infilled circles. **B–D)** Seasonal variability of accumulation rate simulated by p-TOMCAT (black) and the constant annual accumulation rate estimated by ice core dating (green shades). Also displayed on C are snow accumulation rates measured at Summit 2003-2014 AD (purple shades): 'Stakes' = field measurements of snow accumulation at bamboo stakes; 'IceSat' = laser altimetry measurements from ICESat (Zwally et al., 2002). These snow accumulation records have been converted to water equivalent accumulation rate assuming a snow density of 0.34 g cm$^{-3}$. All records are shown with uncertainty bars representing ± 1 σ of inter-annual variability.




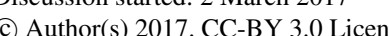

**Figure 7: Time series of Greenland ice core [Na] (green) and p-TOMCAT simulated [Na] in Greenland snow 1991–1999 AD (raw = cyan, smoothed = blue). Simulated [Na] is calculated using the constant annual accumulation rate indicated by the ice core records in Eq. (5). Simulated [Na] calculated using p-TOMCAT accumulation rates is out of phase with ice core data at some sites (see Fig. 5A).**





**Figure 8: Greenland ice core [Na] simulated by p-TOMCAT (1991-1999 AD) compared to Greenland ice core data. A) Annual mean [Na], B) Winter (DJF) mean [Na], C) Summer (JJA) mean [Na] and D) simulated winter (DJF) SISS:OOSS ratio with black crosses marking ice core locations. Simulated [Na] values calculated using the modelled accumulation rate (see Table 2 for alternative values at each ice core site if ice core accumulation rate is used). Note the log scale to both colour bars.**





**Table 1: Key characteristics of Greenland ice core records used.**

| Ice Core | Location | Elevation (m) | Accumulation rate‡ (kg m⁻² yr⁻¹) | Distance to coast (km) | Record end (yr AD) | Reference |
|---|---|---|---|---|---|---|
| Tunu13 | 78°2.09'N, 33°52.80'W | 2105 | 112 | 300 | 2011 | Maselli et al. (2017) |
| NEEM-2011-S1† | 77°26.93'N, 51°03.37'W | 2454 | 203 | 280 | 1997.5 | Sigl et al. (2013) |
| NEEM-2008-S3† | 77°26.93'N, 51°03.37'W | 2454 | 203 | 280 | 2001 | |
| NEEM-2010-20m† | 77°26.93'N, 51°03.37'W | 2454 | 203 | 280 | 2008 | |
| Summit2010 (a.k.a Zoe2) | 72°36.0'N, 38°18.0'W | 3258 | 222 | 530 | 2010 | Maselli et al. (2017) |
| D4 | 71°24.0'N, 43°54.0'W | 2730 | 414 | 320 | 2003 | Banta et al. (2008) |
| D5 | 68°30.0'N, 42°54.0'W | 2468 | 373 | 350 | 1998 | Banta et al. (2008) |
| Das2 | 67°30.0'N, 36°06.0'W | 2936 | 833 | 110 | 2003 | Banta et al. (2008) |
| Das1 * | 66°00.0'N, 44°00.0'W | 2497 | 600 | 200 | 2003 | Banta et al. (2008) |
| ACT10C * | 65°59.93'N, 42°47.0'W | 2299 | 809 | 200 | 2009.5 | |
| ACT3 * | 66°00.0'N, 43°36.0'W | 2508 | 658 | 200 | 2005 | |
| ACT2 *** | 66°00.0'N, 45°12.0'W | 2419 | 372 | 240 | 2004 | Banta et al. (2008) |
| ACT11d *** | 66°28.8'N, 46°18.6'W | 2296 | 339 | 240 | 2011 | |

‡ Water equivalent accumulation rate
† same grid square in p-TOMCAT
* same grid square in p-TOMCAT
5 ** same grid square in p-TOMCAT



**Table 2: Mean sea salt Na concentrations for 1991–1999 AD recorded in ice cores (bold) and simulated by p-TOMCAT: A) [Na] calculated using p-TOMCAT precipitation output in Eq. (5), B) [Na] calculated using the constant annual accumulation rate indicated by the ice core records (Table 1) in Eq. (5).**

| Ice core | | Annual [Na] (ppb) | DJF [Na] (ppb) | JJA [Na] (ppb) | Seasonal cycle [Na] (ppb) ‡ | Rel. seasonal cycle† | DJF SISS: OOSS |
|---|---|---|---|---|---|---|---|
| **Tunu** | | **16** | **24** | **7** | **22** | **5** | |
| | A | 14 | 17 | 9 | 23 | 6 | 0.4 |
| | B | 15 | 26 | 6 | 26 | 7 | 0.3 |
| **NEEM-2008-S3** | | **11** | **25** | **3** | **31** | **12** | |
| | A | 14 | 16 | 7 | 17 | 5 | 1.0 |
| | B | 17 | 17 | 11 | 14 | 3 | 0.8 |
| **Summit** | | **6** | **12** | **1** | **13** | **12** | |
| | A | 11 | 13 | 7 | 16 | 7 | 0.2 |
| | B | 9 | 17 | 3 | 17 | 9 | 0.2 |
| **D4** | | **4** | **8** | **1** | **8** | **7** | |
| | A | 11 | 12 | 7 | 16 | 6 | 0.2 |
| | B | 12 | 18 | 5 | 16 | 6 | 0.1 |
| **D5** | | **8** | **13** | **4** | **10** | **3** | |
| | A | 14 | 16 | 8 | 19 | 5 | 0.2 |
| | B | 23 | 34 | 10 | 30 | 6 | 0.1 |
| **Das2** | | **5** | **12** | **2** | **12** | **16** | |
| | A | 18 | 21 | 13 | 16 | 3 | 0.2 |
| | B | 25 | 37 | 11 | 33 | 6 | 0.2 |
| **Das1*** | | **11** | **23** | **2** | **23** | **14** | |
| | A | 22 | 25 | 13 | 25 | 4 | 0.1 |
| | B | 34 | 50 | 16 | 40 | 4 | 0.1 |
| **ACT10C*** | | **21** | **49** | **4** | **55** | **34** | |
| | A | 22 | 25 | 13 | 25 | 4 | 0.1 |
| | B | 25 | 36 | 12 | 29 | 4 | 0.1 |
| **ACT3*** | | **9** | **19** | **2** | **18** | **12** | |
| | A | 22 | 25 | 13 | 25 | 4 | 0.1 |
| | B | 30 | 44 | 14 | 35 | 4 | 0.1 |
| **ACT2**** | | **8** | **13** | **3** | **10** | **5** | |
| | A | 25 | 24 | 16 | 31 | 4 | 0.1 |
| | B | 50 | 64 | 29 | 43 | 3 | 0.1 |
| **ACT11d**** | | **7** | **10** | **5** | **7** | **3** | |
| | A | 25 | 24 | 16 | 31 | 4 | 0.1 |
| | B | 53 | 68 | 30 | 45 | 3 | 0.1 |

‡ Seasonal cycle is the maximum monthly mean [Na] minus the minimum monthly mean [Na].

5 † Relative seasonal cycle is the maximum monthly mean [Na] divided by minimum monthly mean [Na].

\* same grid square in p-TOMCAT so A values are equal.

\*\* same grid square in p-TOMCAT so A values are equal.