# Peer review of "Sea ice as a source of sea salt aerosol to Greenland ice cores: a model-based study"

_Atmospheric Chemistry and Physics, 2017_

## Referee Comment (RC1) · Anonymous Referee #2 · 22 Mar 2017

*Rhodes et al.* use a chemical transport model to examine the importance of the sea ice source of sea salt aerosol (SISS) relative to the ocean source of sea salt aerosol (OOSS) in the Arctic. They compare their model to observations of sea salt aerosol in the atmosphere and high resolution $Na^+$ measurements in Greenland ice cores. I found this paper very hard to follow and in the end it wasn't clear what was learned from their modeling exercise beyond what others have published.

Because the processes responsible for the emission of SISS into the atmosphere are not well understood, the authors "tune" their model to best match the aerosol observations. In the discussion of all of the different parameters that can be tuned, the manuscript would greatly benefit first from an explicit description of the parameterization for SISS (i.e., show the actual equations, and define all of the variables). Without it, it is very hard to follow the discussion of the model tuning.

It seems however that some of the model tuning has to do with the treatment of aerosol deposition in the model, not just the SISS emission parameterization, the discussion of which is also confusing. Not all of the terms in Equation 1 are defined. What is $^{\alpha C \, PC}L$ and $^{\alpha N \, PN}L$? Is this somehow related to $\alpha_C$, $\alpha_N$, $PN_L$ and $PC_L$? It looks like there must be a mix up of subscripts and superscripts in either the equation or the text. Does the model calculation of dry deposition include gravitational settling of the larger ($r > 4 \, \mu m$) particles? If not, it should. The modeled wet deposition seems to be missing some important processes (Page 13). It's also not clear if the modified snow precipitation directly influences wet deposition, or of the modeled wet deposition uses the "incorrect" precipitation.

I think what is new about this manuscript is the comparison of the model with Greenland ice core $Na^+$ observations. However, this is probably the most ambiguous part of the paper, and it's not clear to me what they learned from this exercise. They are comparing modeled versus observed seasonality, although it seems that the seasonality of ice core $Na^+$ is unclear as it was determined assuming constant snow accumulation rates, which is probably not consistent with reality. Also perhaps the seasonality is not well preserved in the observational record because of factors such as snow redistribution and sastrugi (page 14). In the end it seems that the model shows little skill at simulating the observed seasonality of ice core $Na^+$, although it's questionable whether the "observed" seasonality represents the true seasonality of $Na^+$ deposition to the Greenland ice sheet. The second paragraph of the summary (section 6) I think attempts to articulate what they learned from the model/ice-core observation comparison, but I still cannot figure out what was learned from this exercise. Given that this is the main new contribution of this paper, the paper should be substantially revised to better articulate their scientific contribution.

More minor issues:

Page 2 line 30: The last sentence of this paragraph needs a reference.

Page 5 Lines 16-17: Provide a justification for the choice of 0.3 psu.

Page 5 Line 21 and elsewhere: What does "snow age" mean?  This should be defined.  It's not clear how this should impact SISS.

Page 9 lines 8-9: How was scenario #3 parameterized?  Did you simply reduce salinity by 50%?

Page 9 Line 17: Define NRMSD the first time used.

Page 11 line 1: Unfinished statement.  What are you comparing the model simulations to?

Specify "snow accumulation" instead of just "accumulation" throughout the manuscript.

Page 14 line 24: What is a "Greenland ice core simulation"?  Do you mean model simulation?

Page 15 line 25-26: Be sure to specify that this is for today's climate.  Perhaps it would be different in a different climate.

Figure S1 should be in the main text.

When Figure 3 is presented in the text, it is not yet clear what your "base case" simulation is, which I think is what the blue line is in the figure.  This information should be presented in order.

Figure 7: What are the yellow and other 3 green colors?  The acronyms should be restated in the figure caption.

Figure 8: The model-observation comparison appears good here probably because of the large (2 order-of-magnitude) range in the color bar.  The observations themselves cover a much smaller range, so the color bar should be scaled according to the range of the observations.  Also I'm not sure this is the appropriate figure type to show because of the uncertainties in the SISS parameterizations.  It would be best to have a figure that communicates the full model range using all of your sensitivity simulations.

---

## Referee Comment (RC2) · Anonymous Referee #1 · 2 Apr 2017

This manuscript uses the p-TOMCAT CTM to examine the role of sea ice sources of sea salt on atmospheric concentrations of Na+ in the Arctic and in ice core measurements in Greenland. The authors propose that the model including sea ice sources can reproduce the winter maximum of Na+ in the atmosphere and in ice core records. Furthermore, the authors show that the model reproduces some of the interannual variability at Summit, which could provide a way to use Na+ concentrations in ice cores as a proxy for sea ice extent.

This study builds on previous work by the authors in developing a parameterization for blowing snow and applying it to examine halogen activation as well as sea salt emissions over polar regions (Yang et al., 2008; Levine et al., 2014; Legrand et al., 2016). The authors further refine their assumptions on snow salinity and snow age.

General Comments

I have 3 main areas of concern regarding this study. The first is that the description of the different tuning parameters is confusing in the manuscript. In particular, after the sensitivities studies it is unclear what the final choice is for the standard simulation used in the figures. The second area of concern is that many parameters are changed compared to the 3 previous simulations using p-TOMCAT (Yang et al., 2010; Levine et al., 2014; Legrand et al., 2016). It appears that depending on the problem at hand (Arctic vs Antarctic sea salt; sea salt vs bromine), different parameters are tuned to different values. This again tends to be confusing. It would be very useful to the reader to get a sense of what the different assumptions were in the different studies and how they affect total emissions of SISS. Furthermore, OOSS formulations also seem to be different in each of these papers. Finally, the comparison between Greenland ice core measurements and the model show that the open ocean source of sea salt is sufficient to explain the observed seasonal cycle at Na+, thus the authors' conclusion that Na+ concentrations in ice cores is influenced by sea ice sources is not supported by their comparison.

These comments and other specific comments are detailed in the next sections.

Specific Comments

1) Snow salinity. After reading section 2.3.2, it is unclear to me what salinity is used for Arctic snow on sea ice. The authors mention the BLOWSEA project with 0.3 psu for Antarctic snow salinity. Is that the value used in the standard model shown in Figure 3? In section 3.3.1 (page 8), the authors mention a sensitivity simulation with 2-fold and 3-fold salinity. What is that with respect to? 0.3 psu? This is confusing, and it would be clearer to directly specify the actual numerical value of the salinity used. Is the 2-fold salinity 0.6 psu and 3-fold salinity 0.9 psu? Which one is used in Figure 4? I suggest that the author discuss the different salinities used in section 2.3.2 and then refer to them in the sensitivity studies

2) Sea salt emissions. Can the authors compare their emissions (in TgNa/yr) for both OOSS and SISS to Huang and Jaegle (2017)?

3) This is the fourth paper using the Yang et al. (2008) blowing snow parameterization in P-TOMCAT (Yang et al., 2010; Levine et al., 2014; Legrand et al., 2016). In each of these papers different assumptions are made in terms of OOSS source functions, as well as blowing snow parameters (salinity, snow age, gustiness, etc...). It would be useful to discuss the overall impact of these different assumptions on emissions. In particular, I suggest adding a table that lists Arctic and Antarctic emissions for Na for both OOSS and SISS (this could be use to address my comment 2) above). This table should also include mean surface concentrations or tropospheric burdens of Na.

4) Snow age. Page 5, line 22. The choice of 24 hour snow age seems arbitrary, especially as a previous study with the same model used a snow age of 5 days. A better justification of this value would be to use the meteorological fields to infer a mean time between snow precipitation over the Arctic.

5) Comparison to atmospheric observations (Figure 3). The observations at the different sites are for different time periods (Alert: 1990-1995; Summit: 2003-2006; Barrow: 1997-2000; Zeppelin: 1993-1999; Villum: 2001, 2002, 2008-2013) but the model simulation is the average for 1991-1999, which in the case of the Greenland sites doesn't overlap with the observations. For the other sites, there is some overlap, but the model years are not selected to match the observation years. Given the large interannual variability in Na observations (and in the simulations) can the authors justify this approach? I suggest that at a minimum the authors select the model years that match the observations for Alert, Barrow, Zeppelin. Extending their simulation by a few years would also allow them to have a more rigorous comparison to the Greenland sites.

6) Section 3.3 and figure 4. The sensitivity studies shown in Figure 4 are conducted for a single year (1997), while the observations are for multiple years – at least this seems to be the case based on Figure S4. How representative is 1997 compared to

the 1991-1999 simulations? At some sites, such as Villum (Figure S4) there appears to be significant differences between 1997 and the 1991-1999 average. Is panel A in Figure 4 for 1997 only or for 1991-1999 (corresponding to Figure 2)? Based on this single year simulation, my understanding that authors choose the option with multi-year sea ice emissions decreased by 50% (panel C) for subsequent simulations (page 9, line 15). The authors should justify this. If this is the simulation they choose, it should be the one they show in Figure 3. To clarify the assumptions for the various simulations, the authors should include a table in the supplementary material with the actual assumptions that are made. For example what salinity (over what sea ice) and snow age are used in Figure 4E?

7) Page 10 line 15. Do the authors have any potential explanations for why the observations at Barrow are reproduced by the SISS simulation during the first part of the year, but not the second part? Are the meteorological conditions (windspeed) not captured as well?

8) Seasonal variability of Na in ice cores. The authors compare the p-TOMCAT simulation to ice core observations over Greenland, finding that the model captures the observed seasonality with a winter maximum (section 4.3.2). Figure 5 shows that this seasonality is mostly due to the open ocean SS aerosol (dashed red line), with little influence from the sea ice SS sources. This is contrast to the open ocean (OOSS) simulation of atmospheric Na at ground sites in the Arctic (Figure 3). Can the authors explain the reason for this different modeled seasonality in the atmosphere and in ice cores for the OOSS simulation? Also the comparison between p-TOMCAT and ice core measurements is a little difficult to follow as different sites are shown in different figures. For example, Tunu is missing from figure 5, but is shown in Figure 7. I suggest that the authors add Tunu in Figure 5, especially as it appears that the modeled influence of sea ice sources might be large at this site.

9) Section 5. Based on the comparison shown in Figure 5, it seems that the sea ice sources do not really lead to a better simulation of the ice core measurements.

At most sites the influence of sea ice sources is small. The largest modeled sea ice influence is at the NEEM site, where the model does not capture the observed seasonal cycle. Thus this comparison is inconclusive in terms of the role of a sea ice source in influencing ice core measurements.

Technical corrections

- Page 2 line 24: add "the" before "principal source of sea salt"

- Page 2 line 30. Please add a reference for these field-based observations in the Weddell Sea.

- Page 3 line 27. "age range" is a strange term. Do the authors mean "year of collection"?

- Page 8 line 20-22. This sentence is confusing. The Weddell sea salinity (0.3 psu) multiplied by two is 0.6 psu, while this sentence implies it is 0.12 psu. The Mundy observations of 0.1 psu of surface snow over the central Canadian Arctic thus imply that the salinity used by this study (0.6 psu?) is too large.

- Page 9 line 10. Can the authors be more specific about the region chosen to calculate these emissions?

- Page 15 line 16. "SISS contributes to the winter maxima observed in all the ice cores, but that in some cases, OOSS alone can produce winter maxima and summer minima in sea salt in ice cores" There is no evidence of this in the manuscript. Figure 5 shows that OOSS reproduces the observed seasonal cycle at all sites except for NEEM. At NEEM, adding the SISS source doesn't lead to a better simulation.

---

## Author Comment (AC1) · 9 Jun 2017

**Rhodes et al. reply to Anonymous Reviewer #1**

We thank the reviewer for their thoughtful comments and suggestions. We address each one directly below and outline changes that will be made to a revised manuscript.

Specific Comments
1) Snow salinity. After reading section 2.3.2, it is unclear to me what salinity is used for Arctic snow on sea ice. The authors mention the BLOWSEA project with 0.3 psu for Antarctic snow salinity. Is that the value used in the standard model shown in Figure 3?
In section 3.3.1 (page 8), the authors mention a sensitivity simulation with 2-fold and 3-fold salinity. What is that with respect to? 0.3 psu? This is confusing, and it would be clearer to directly specify the actual numerical value of the salinity used. Is the 2-fold salinity 0.6 psu and 3-fold salinity 0.9 psu? Which one is used in Figure 4? I suggest that the author discuss the different salinities used in section 2.3.2 and then refer to them in the sensitivity studies.
0.3 psu is the mean value of the salinity distribution of snow-on-sea-ice measurements from the top 10 cm of snow collected in the Weddell Sea, Antarctica. For Fig. 3, Fig. 4C and others (now called the base simulation) we use double the values of this salinity distribution for snow on Arctic sea ice. The mean salinity used in simulations is therefore 0.6 psu and Fig. 4D (3x salinity) uses a mean value of 0.9 psu. Labels on Fig. 4 have been changed to clarify this. The wording of Section 2.3.2 is confusing and it will be re-written in a revised manuscript.

2) Sea salt emissions. Can the authors compare their emissions (in TgNa/yr) for both OOSS and SISS to Huang and Jaegle (2017)?
Yes, this is something we have done for our own information and can include the results in an additional table (below). A comparison between GEOS-chem and p-TOMCAT will be included in section 3. Essentially the table suggests that both models simulate OOSS emission, transport and deposition in a similar way. p-TOMCAT has higher rates of emission and deposition of larger SISS particles, causing a lower burden and lifetime in the Arctic region. This difference arises from Huang and Jaeglé's assumption that each blowing snow particle produces five sea salt aerosol, whereas in p-TOMCAT one aerosol is produced from each blowing snow particle. There is no evidence to determine with parameterisation is more realistic.

**Table x: Comparison between Arctic (>60ºN) sea salt aerosol budgets simulated by this study (black) and by Huang and Jaeglé (2017) (blue) for 2005 AD. For comparison with Huang and Jaeglé (2017) all values refer to mass of total sea salt aerosol (Na mass multiplied by 0.326). Please see footnotes for definitions of each term.**

| | OOSS | | | SISS | | |
|---|---|---|---|---|---|---|
| This study | 0.01<dryr<= 0.57 μm | 0.57<dryr<= 4.5 μm | Total | 0.01<dryr<= 0.57 μm | 0.57<dryr<= 4.5 μm | Total |
| Huang and Jaeglé 2017 | 0.01–0.5 μm | 0.5–4 μm | Total | 0.01–0.5 μm | 0.5–4 μm | Total |
| Emission rate (Tg yr⁻¹) | 0.69 | 24 | 25 | 0.41 | 8.4 | 8.8 |
| | 0.78 | 29 | 30 | 1.0 | 1.6 | 2.6 |
| Burden (Gg) | 3.0 | 24 | 27 | 1.6 | 1.9 | 3.5 |
| | 12 | 32 | 45 | 14 | 3.3 | 17 |
| Surface concentration (μg m⁻³) | 0.07 | 0.50 | 0.57 | 0.11 | 0.13 | 0.24 |
| | 0.19 | 1.0 | 1.2 | 0.4 | 0.17 | 0.57 |
| Deposition rate (Tg yr⁻¹) | 0.85 | 25 | 26 | 0.34 | 8.3 | 8.6 |
| | 1.3 | 33 | 34 | 0.78 | 1.7 | 2.4 |
| | 1.3 | 0.35 | 0.38 | 1.7 | 0.08 | 0.15 |

| | | | | | | |
|---|---|---|---|---|---|---|
| Lifetime in Arctic region (days) | 3.3 | 0.35 | 0.48 | 6.6 | 0.73 | 2.6 |

Emission rate: Mean rate of sea salt aerosol emission across the Arctic for 2005 AD; Burden: Annual mean total mass of sea salt aerosol present in the Arctic atmosphere (entire column) in 2005 AD. Surface concentration: Mean concentration of sea salt aerosol across the Arctic region in the surface layer of the atmosphere (as defined by model, in p-TOMCAT ≈ 46-72 m height) in 2005 AD; Deposition rate: Mean rate of sea salt aerosol deposition (wet and dry removal) across the Arctic for 2005 AD; Lifetime: Lifetime of sea salt aerosol in the Arctic region calculated as Burden (Tg) /Deposition Rate (Tg yr$^{-1}$). This value will be influenced by import or export of sea salt aerosol to/from Arctic region (which must be occurring when Emission Rate ≠ Deposition Rate.

3) This is the fourth paper using the Yang et al. (2008) blowing snow parameterization in P-TOMCAT (Yang et al., 2010; Levine et al., 2014; Legrand et al., 2016). In each of these papers different assumptions are made in terms of OOSS source functions, as well as blowing snow parameters (salinity, snow age, gustiness, etc...). It would be useful to discuss the overall impact of these different assumptions on emissions. In particular, I suggest adding a table that lists Arctic and Antarctic emissions for Na for both OOSS and SISS (this could be use to address my comment 2) above). This table should also include mean surface concentrations or tropospheric burdens of Na.

Good idea, OOSS and SISS emissions and tropospheric burdens can be added to the new table, as above.

Additionally, we have run several tests for the year 1997 to quantify the impact of choices made in the OOSS parameterisation. Capital letters refer to new figure below: B) OOSS emissions with no SST dependence (as Levine et al., 2014 and Legrand et al., 2016); C) Gustiness factor of 1.17 used to increase surface winds speeds involved in OOSS and SISS emissions (as Levine et al., 2014). Note this change also impacts dry deposition; and D) f(SST) = 0.25 when SST <5°C & no OOSS emissions in grid square if < 50% water (modifications of Huang and Jaeglé (2017) made to Jaeglé et al. (2011) scheme). The results will be included in a supplementary figure (below).

[Figure]

We have also run a 5$^{th}$ variation of the SISS emissions parameterisation where the snow age parameter is set to zero/neglected. Results will be included on Figure 4 (below) and Figure S4.

[Figure]

**Figure 4: Sensitivity of p-TOMCAT Na aerosol simulations for 1997 AD at 5 Arctic locations to parameters associated with SISS emissions via blowing snow. Each panel (A-E) displays the mean difference between monthly (not including July-September) model results and observations (ΔNa) for each site. Positive [negative] values indicate that p-TOMCAT over- [under-] estimates aerosol Na concentration. The normalised root mean square difference (NRMSD) between model simulations and aerosol data is calculated for each of the 5 sites and the mean NRMSD across all 5 sites is displayed on each subplot. Plots of simulated monthly Na concentration at each site, under each scenario, are displayed in Fig. S5. In the base simulation, SISS emissions are reduced by 50% over multi-year sea ice relative to 1st year sea ice, mean snow salinity is 0.6 psu, and snow age is 24 hr.**

We note that there are major differences in the precipitation schemes (and therefore wet deposition) between Yang et al., (2008), Levine et al., (2014), and Legrand et al., (2016). Changes to the OOSS and SISS emissions are not responsible for all the difference between the studies. This study uses the same precipitation scheme as Legrand (2016), which is the most accurate as it is forced towards observational data.

4) Snow age. Page 5, line 22. The choice of 24 hour snow age seems arbitrary, especially as a previous study with the same model used a snow age of 5 days. A better justification of this value would be to use the meteorological fields to infer a mean time between snow precipitation over the Arctic.

The snow age parameter was originally included in the SISS emission parameterisation of Yang et al. (2008) to reflect how the sintering together of snow flakes/crystals over time may cause them to be less likely to be lofted up during blowing snow events. Although the precipitation amount in the Arctic region is simulated well in p-TOMCAT at the monthly or annual scale, the frequency and/or intensity of precipitation events may be less accurate. For this reason, one value of snow age was adopted for the entire Arctic or Antarctic by Levine et al. (2014). The snow age parameter has therefore become more of a tuning tool with little physical basis. Levine et al. (2014) used a value of 5 days, which effectively counteracted their high snow salinity (relative to this study). The impact of neglecting the snow age parameter will be reported (see answer to 3).

5) Comparison to atmospheric observations (Figure 3). The observations at the different sites are for different time periods ... but the model simulation is the average for 1991-1999, which in the case of the Greenland sites doesn't overlap with the observations. For the other sites, there is some overlap, but the model years are not selected to match the observation years. Given the large interannual variability in Na observations (and in the simulations) can the authors justify this approach? I suggest that at a minimum the authors select the model years that match the observations for Alert, Barrow, Zeppelin. Extending their simulation by a few years would also allow them to have a more rigorous comparison to the Greenland sites.

Great suggestion. A revised Fig. 3 (below) shows only overlapping years of model simulation and aerosol data for all sites (where possible). We have been able to extend the model run to 2006 AD. Years in black text indicate aerosol data and year range in red text refers to the model simulation.

[Figure]

6) Section 3.3 and figure 4. The sensitivity studies shown in Figure 4 are conducted for a single year (1997), while the observations are for multiple years – at least this seems to be the case based on Figure S4. How representative is 1997 compared to the 1991-1999 simulations? At some sites, such as Villum (Figure S4) there appears to be significant differences between 1997 and the 1991-1999 average.

1997 was chosen for the sensitivity tests because, across the 5 aerosol sites, it is close to the 1991-1999 mean (mean value of 5 sites NRMSDs between yearly results and 1991-1999 mean is 45%, at Villum only it is 37%). 1992 is slightly closer to the 1991-1999 mean (37% NRMSD), but 1997 overlaps better with aerosol observations.

Is panel A in Figure 4 for 1997 only or for 1991-1999 (corresponding to Figure 2)? Based on this single year simulation, my understanding that authors choose the option with multiyear sea ice emissions decreased by 50% (panel C) for subsequent simulations (page 9, line 15). The authors should justify this. If this is the simulation they choose, it should be the one they show in Figure 3. To clarify the assumptions for the various simulations, the authors should include a table in the supplementary material with the actual assumptions that are made. For example what salinity (over what sea ice) and snow age are used in Figure 4E?

All panels on Figure 4 represent results for 1997 only (equivalent to Fig. S4) – the caption will be altered (as above). The same settings are used to produce Fig. 4 C for 1997 and Fig. 3 for 1991-1999. We agree that this could be much clearer, particularly in identifying Fig. 4 C as the base simulation. As discussed above, the salinity and snow age values have been added to Figure 4 (please see Figure 4 above).

7) Page 10 line 15. Do the authors have any potential explanations for why the observations at Barrow are reproduced by the SISS simulation during the first part of the year, but not the second part? Are the meteorological conditions (windspeed) not captured as well?

We speculate that the snow salinity on the sea ice surface should vary seasonally with the cycle of sea ice formation and melt. Maybe in the Autumn and early Winter when the sea ice is still forming and holds more brine, the surface snow is saltier, possibly causing saltier SISS aerosol? More observations are needed.
We don't know of any reason to doubt the ERA interim wind data at this location.

8) Seasonal variability of Na in ice cores. The authors compare the p-TOMCAT simulation to ice core observations over Greenland, finding that the model captures the observed seasonality with a winter maximum (section 4.3.2). Figure 5 shows that this seasonality is mostly due to the open ocean SS aerosol (dashed red line), with little influence from the sea ice SS sources. This is contrast to the open ocean (OOSS) simulation of atmospheric Na at ground sites in the Arctic (Figure 3). Can the authors explain the reason for this different modeled seasonality in the atmosphere and in ice cores for the OOSS simulation?

Most of the Arctic aerosol sites on Fig. 3 show little seasonality in OOSS but the model results for Summit do suggest seasonality in aerosol OOSS at that location. This is then in agreement with the model results for Greenland ice cores (including at Summit) on Fig. 5. Because the Greenland ice cores are located further south than the aerosol sampling sites (excepting Summit, Fig. 1), they are influenced more strongly by OOSS sources. The reason for the simulated OOSS seasonality at Greenland ice core sites is difficult to isolate. It may be related to the seasonality of precipitation in the model, which controls wet deposition occurrence.

Also the comparison between p-TOMCAT and ice core measurements is a little difficult to follow as different sites are shown in different figures. For example, Tunu is missing from figure 5, but is shown in Figure 7. I suggest that the authors add Tunu in Figure 5, especially as it appears that the modeled influence of sea ice sources might be large at this site.

Tunu was not originally included on Fig. 5 because it is a relatively low accumulation site (~11 cm water/year), meaning that the seasonal signal in [Na] is not as well-defined as at other sites, particularly in deeper (older) sections of the core. However, for the 1990s, the data look good and Tunu is the most northerly ice core with a relatively high proportion of SISS so the D5 ice core will be replaced with Tunu on Fig. 5. Please see revised figure below.

[Figure]

9) Section 5. Based on the comparison shown in Figure 5, it seems that the sea ice sources do not really lead to a better simulation of the ice core measurements. At most sites the influence of sea ice sources is small. The largest modeled sea ice influence is at the NEEM site, where the model does not capture the observed seasonal cycle. Thus this comparison is inconclusive in terms of the role of a sea ice source in influencing ice core measurements.

The first sentence of section 5 will be re-phrased to emphasize that the importance of SISS to the ice core [Na] budget differs geographically, SISS does not make an "important contribution" in southern Greenland. At NEEM the model captures the seasonality (max in winter, min in summer) in [Na], again the ice core [Na] peak might occur in early spring but be fixed to Jan 1st by the ice core dating technique. We believe the other statements in section 5 are sound and do not overstate the influence of SISS on Greenland ice core [Na].

Technical corrections:

- Page 8 line 20-22. This sentence is confusing. The Weddell sea salinity (0.3 psu) multiplied by two is 0.6 psu, while this sentence implies it is 0.12 psu. The Mundy observations of 0.1 psu of surface snow over the central Canadian Arctic thus imply that the salinity used by this study (0.6 psu?) is too large.

We agree, this is confusing. The *median* Weddell sea salinity x2 is 0.12 psu, which is similar to the *mean* salinity reported by Mundy for surface snow (0.11 psu).

- Page 15 line 16. "SISS contributes to the winter maxima observed in all the ice cores, but that in some cases, OOSS alone can produce winter maxima and summer minima in sea salt in ice cores" There is no evidence of this in the manuscript. Figure 5 shows that OOSS reproduces the observed seasonal cycle at all sites except for NEEM. At NEEM, adding the SISS source doesn't lead to a better simulation.

In the text the quotation above starts with the qualifying statement "Our simulations...suggest that...". Fig. 5 shows simulations that indicate SISS contributes to the winter [Na] maxima at ice core sites. Because adding the additional SISS input in winter months causes the simulated values to increase further beyond the measured ice core values doesn't mean that the SISS input does not occur or is not important. At most sites the summer OOSS Na concentrations are higher than the ice core measurements, suggesting that deposition of OOSS over Greenland is over-estimated by p-TOMCAT.

Other technical corrections will be addressed in a revised manuscript.

---

## Author Comment (AC2) · 9 Jun 2017

**Rhodes et al. reply to Anonymous Reviewer #2**

We thank the reviewer for their comments and address each one directly below.

Rhodes et al. use a chemical transport model to examine the importance of the sea ice source of sea salt aerosol (SISS) relative to the ocean source of sea salt aerosol (OOSS) in the Arctic. They compare their model to observations of sea salt aerosol in the atmosphere and high resolution Na+ measurements in Greenland ice cores. I found this paper very hard to follow and in the end it wasn't clear what was learned from their modeling exercise beyond what others have published.

As is stated several times in the manuscript (including the abstract and section 6 summary) this is the first study to utilise a chemical transport model to simulate the sea salt concentrations of snow, therefore enabling direct comparison with ice core records. Furthermore, we show that Na concentrations can be simulated to within a factor of 2. As alluded to in the abstract and detailed in the text, this is the first manuscript of a wider study into the viability of ice core sea salt as an ice core proxy. It will be followed by an investigation into the factors driving signal variability (pg. 15, line 23) and studies using paleoclimate boundary conditions (pg. 15, line 6). We trust that our manuscript will be easier to follow once the suggestions provided by reviewers are implemented.

Because the processes responsible for the emission of SISS into the atmosphere are not well understood, the authors "tune" their model to best match the aerosol observations. In the discussion of all of the different parameters that can be tuned, the manuscript would greatly benefit first from an explicit description of the parameterization for SISS (i.e., show the actual equations, and define all of the variables). Without it, it is very hard to follow the discussion of the model tuning.

We choose not to repeat the equations for SISS emission of Yang et al. (2008) because we do not introduce any new parameterisation or variables. We refer to specific equation numbers in the text so that the reader can refer back to Yang et al. (2008).

It seems however that some of the model tuning has to do with the treatment of aerosol deposition in the model, not just the SISS emission parameterization, the discussion of which is also confusing.

The parameterisations of sea salt deposition have not been altered since Levine et al.'s (2014) work. What we describe in section 2.3.3 is how the amount of sea salt *deposited* at each time step is calculated. Previously, only the amount *remaining in the atmosphere* after deposition was calculated. We do this in order to make a direct comparison with ice core sea salt concentrations (rather than atmospheric aerosol concentrations) (pg. 6, line 9).

Please see our reply to Reviewer 1 for suggested changes that will make the SISS emission parameterisation easier to follow.

Not all of the terms in Equation 1 are defined. What is $\alpha C$ PCL and $\alpha N$ PNL? Is this somehow related to $\alpha C$, $\alpha N$, PNL and PCL? It looks like there must be a mix up of subscripts and superscripts in either the equation or the text.

Yes, this is a typo mixing up subscripts and superscripts that will be rectified.

Does the model calculation of dry deposition include gravitational settling of the larger ($r > 4$ $\mu$m) particles? If not, it should.

Yes, see Pg. 6 line 8.

The modeled wet deposition seems to be missing some important processes (Page 13). It's also not clear if the modified snow precipitation directly influences wet deposition, or of the modeled wet deposition uses the "incorrect" precipitation.

The wet deposition scheme does have limitations, as we acknowledge (Pg. 13, line 20). The wet deposition code uses the model-generated precipitation (black line on Fig. 6 B-D) and this can be stated clearly in text.

I think what is new about this manuscript is the comparison of the model with Greenland ice core Na+

observations. However, this is probably the most ambiguous part of the paper, and it's not clear to me what they learned from this exercise. They are comparing modeled versus observed seasonality, although it seems that the seasonality of ice core Na+ is unclear as it was determined assuming constant snow accumulation rates, which is probably not consistent with reality. Also perhaps the seasonality is not well preserved in the observational record because of factors such as snow redistribution (page 14).

The seasonality in ice core Na is significant in all the Greenland ice cores shown on Fig. 5 (see green lines and uncertainty bars that denote interannual variability). This seasonality is preserved even though processes such as snow redistribution have likely impacted the ice core records.

What may be uncertain is the *monthly timing* of peak [Na] because when an ice core is dated by counting of annual layers in chemistry records [Na] is often assumed to peak Jan 1st. Or more accurately, the ratio of non-sea-salt sulphur (nssS) (mostly from sulphate) to Na is used and the minimum is dated as Jan 1st. The timing of the [Na] may therefore be artificially fixed as Jan 1st, when it could vary by few months either side. Support for the winter timing of peak [Na] comes from sea salt measurements of Arctic aerosol (Fig. 3) and fresh snow e.g., at Summit (Fig. S5). Figure S5 now includes nssS:Na measured in snow at Summit.

In the end it seems that the model shows little skill at simulating the observed seasonality of ice core Na+...

The model simulates summer minima in [Na] and maxima in either winter or spring. This is similar to the observed ice core [Na] seasonality, especially when we bear in mind that the Na peaks may have been used as winter (Jan 1st) markers in ice core dating, thereby artificially fixing the timing of maximum [Na]. The model simulates the relative and absolute amplitudes of the seasonal cycle reasonably well (pg. 13, line 28 & Table 2).

The second paragraph of the summary (section 6) I think attempts to articulate what they learned from the model/ice-core observation comparison, but I still cannot figure out what was learned from this exercise. Given that this is the main new contribution of this paper, the paper should be substantially revised to better articulate their scientific contribution.

We thank the reviewer for their suggestion and will work on this in a revised manuscript.

Page 2 line 30: The last sentence of this paragraph needs a reference.

A publication discussing these results in detail is currently being prepared. Once published, the data set will be available online at the NERC Polar Data Centre. A DOI is currently being generated, where metadata will be made available. The DOI will included in the revised paper so researchers can access the data easily in the future.

Page 5 Lines 16-17: Provide a justification for the choice of 0.3 psu.

0.3 psu is the mean of the salinity distribution of samples from the top 10 cm of the snow pack on sea ice in the Weddell Sea. This could be made clearer in the text.

Page 5 Line 21 and elsewhere: What does "snow age" mean? This should be defined. It's not clear how this should impact SISS.

The snow age parameter is defined in Box et al. [2004] and introduced by Yang et al. [2008] to represent the efficiency of aged snow in SSA production. The snow age parameter taken by Yang et al. (2008) to be the number of hours since the last snow event. We agree that it should be defined here too. The idea is that fresher snow is more easily lofted up because the individual grains/flakes haven't had time to sinter together. So, a higher snow age decreases the SISS emissions. Please also see reply to Reviewer #1.

Page 9 lines 8-9: How was scenario #3 parameterized? Did you simply reduce salinity by 50%?

In scenario #3 the area of multi-year sea ice in each grid cell used in SISS emission calculation is halved. This has the effect of halving the SISS emissions from the multi-year ice in that grid cell. It is not precisely the same as reducing the snow salinity by 50%. Sea ice area is only halved for this calculation, it does not result in additional area of open ocean. The text has been altered to clarify this.

Page 9 Line 17: Define NRMSD the first time used.

Defined on Pg. 7 line 14.

Page 11 line 1: Unfinished statement. What are you comparing the model simulations to?

This sentence will be re-worded.

Specify "snow accumulation" instead of just "accumulation" throughout the manuscript.

This can be done.

Page 14 line 24: What is a "Greenland ice core simulation"? Do you mean model simulation?

This sentence will be re-worded.

Page 15 line 25-26: Be sure to specify that this is for today's climate. Perhaps it would be different in a different climate.

The sentence can be changed from "in the Late Holocene" to "under present day conditions".

Figure S1 should be in the main text.

We disagree. Fig. S1 presents comparisons of aerosol Na measurements and simulations at low latitudes and is not integral to the manuscript or its conclusions.

When Figure 3 is presented in the text, it is not yet clear what your "base case" simulation is, which I think is what the blue line is in the figure. This information should be presented in order.

Yes, we agree, this is confusing and it will be changed. An additional small table will summarise the parameters used in the base simulation and this table will be referred to early on.

Figure 7: What are the yellow and other 3 green colors? The acronyms should be restated in the figure caption.

The caption of Figure 7 will be changed. The other colours are other ice core records located within the same grid square in p-TOMCAT.

Figure 8: The model-observation comparison appears good here probably because of the large (2 order of-magnitude) range in the color bar. The observations themselves cover a much smaller range, so the color bar should be scaled according to the range of the observations. Also I'm not sure this is the appropriate figure type to show because of the uncertainties in the SISS parameterizations. It would be best to have a figure that communicates the full model range using all of your sensitivity simulations.

Both scales are log scales to incorporate the wide range of values. Using a linear scale provided no useful information to the reader. The log scale means the variability at low values (most of the observations) is well represented in the colour scale. Readers can also consult Table 2 for the [Na] values of both model and ice core data. We understand the reviewers concerns about the sensitivity of our SISS:OOSS results (Fig. 8D) to the SISS parameterisation. We have designed two experiments intended to produce extreme SISS:OOSS in order to provide a range of possible SISS:OOSS ratios. These experiments are currently running and results will be included in the supplement.

References:

Box, J. E., D. H. Bromwich, and L.-S. Bai (2004), Greenland ice sheet surface mass balance 1991 – 2000: Application of Polar MM5 mesoscale model and in situ data, J. Geophys. Res., 109, D16105, doi:10.1029/2003JD004451.

---

## Author Response (AR1)

Here we include a point-by-point reply to the comments of two anonymous reviewers. The line numbers refer to the revised, marked-up, manuscript included in this file. We again extend our thanks to the reviewers who have helped improve this manuscript.

**5 Reviewer 1**

Specific Comments

1) Snow salinity. After reading section 2.3.2, it is unclear to me what salinity is used for Arctic snow on sea ice. The authors mention the BLOWSEA project with 0.3 psu for Antarctic snow salinity. Is that the value used in the standard model shown in Figure 3?

- 10 In section 3.3.1 (page 8), the authors mention a sensitivity simulation with 2-fold and 3-fold salinity. What is that with respect to? 0.3 psu? This is confusing, and it would be clearer to directly specify the actual numerical value of the salinity used. Is the 2-fold salinity 0.6 psu and 3-fold salinity 0.9 psu? Which one is used in Figure 4? I suggest that the author discuss the different salinities used in section 2.3.2 and then refer to them in the sensitivity studies.
- 0.3 psu is the mean value of the salinity distribution of snow-on-sea-ice measurements from the top 10 cm of snow collected in the Weddell Sea, Antarctica. For Fig. 3, Fig. 4C and others (now called the base simulation) we use double the values of this salinity distribution for snow on Arctic sea ice. The mean salinity used in simulations is therefore 0.6 psu and Fig. 4D (3x salinity) uses a mean value of 0.9 psu. Labels on Fig. 4 have been changed to clarify this. Section 2.3.2 has been re-written to make this clear. A new table (Table 1) now summarises the parameters used in the base simulation.
- 20

2) Sea salt emissions. Can the authors compare their emissions (in TgNa/yr) for both OOSS and SISS to Huang and Jaegle (2017)?

We include these data in an additional table (Table 3). A comparison between GEOS-Chem and p-TOMCAT is be included in section 3.4. Essentially Table 3 suggests that both models simulate OOSS emission, transport and

- 25 deposition in a similar way. p-TOMCAT has higher rates of emission and deposition of larger SISS particles, causing a lower burden and lifetime in the Arctic region. This difference arises from Huang and Jaeglé's assumption that each blowing snow particle produces five sea salt aerosol, whereas in p-TOMCAT one aerosol is produced from each blowing snow particle.
- 30 3) This is the fourth paper using the Yang et al. (2008) blowing snow parameterization in P-TOMCAT (Yang et al., 2010; Levine et al., 2014; Legrand et al., 2016). In each of these papers different assumptions are made in terms of OOSS source functions, as well as blowing snow parameters (salinity, snow age, gustiness, etc...). It would be useful to discuss the overall

impact of these different assumptions on emissions. In particular, I suggest adding a table that lists Arctic and Antarctic emissions for Na for both OOSS and SISS (this could be use to address my comment 2) above). This table should also include mean surface concentrations or tropospheric burdens of Na.

OOSS and SISS emissions and tropospheric burdens are included in Table 3. Additionally, we ran several tests
for the year 1997 to quantify the impact of choices made in the OOSS parameterisation (Fig. S4, below). Capital letters refer to Figure S4: B) OOSS emissions with no SST dependence (as Levine et al., 2014 and Legrand et al., 2016); C) Gustiness factor of 1.17 used to increase surface winds speeds involved in OOSS and SISS emissions (as Levine et al., 2014). Note this change also impacts dry deposition; and D) f(SST) = 0.25 when SST <5°C & no OOSS emissions in grid square if < 50% water (modifications of Huang and Jaeglé (2017) made to Jaeglé et al. (2011) scheme).</li>

We also ran a 5th variation of the SISS emissions parameterisation where the snow age parameter is set to 15 zero/neglected. Results are included on Figure 4.

We note that there are major differences in the precipitation schemes (and therefore wet deposition) between Yang et al., (2008), Levine et al., (2014), and Legrand et al., (2016). Changes to the OOSS and SISS emissions

are not responsible for all the difference between the studies. This study uses the same precipitation scheme as Legrand (2016), which is the most accurate as it is forced towards observational data.

4) Snow age. Page 5, line 22. The choice of 24 hour snow age seems arbitrary, especially as a previous study with the same5 model used a snow age of 5 days. A better justification of this value would be to use the meteorological fields to infer a mean time between snow precipitation over the Arctic.

The snow age parameter was originally included in the SISS emission parameterisation of Yang et al. (2008) to reflect how the sintering together of snow flakes/crystals over time may cause them to be less likely to be lofted up during blowing snow events. Although the precipitation amount in the Arctic region is simulated well in p-

- 10 TOMCAT at the monthly or annual scale, the frequency and/or intensity of precipitation events may be less accurate. For this reason, one value of snow age was adopted for the entire Arctic or Antarctic by Levine et al. (2014). The snow age parameter has therefore become more of a tuning tool with little physical basis (see pg. 5, lines 18-26).
- 15 5) Comparison to atmospheric observations (Figure 3). The observations at the different sites are for different time periods ... but the model simulation is the average for 1991-1999, which in the case of the Greenland sites doesn't overlap with the observations. For the other sites, there is some overlap, but the model years are not selected to match the observation years. Given the large interannual variability in Na observations (and in the simulations) can the authors justify this approach? I suggest that at a minimum the authors select the model years that match the observations for Alert, Barrow, Zeppelin.
- 20 Extending their simulation by a few years would also allow them to have a more rigorous comparison to the Greenland sites. Figure 3 has been so that only overlapping years of model simulation and aerosol data are used for all sites (where possible). We have been able to extend the model run to 2006 AD.

6) Section 3.3 and figure 4. The sensitivity studies shown in Figure 4 are conducted for a single year (1997), while the
25 observations are for multiple years – at least this seems to be the case based on Figure S4. How representative is 1997 compared to the 1991-1999 simulations? At some sites, such as Villum (Figure S4) there appears to be significant differences between 1997 and the 1991-1999 average.

1997 was chosen for the sensitivity tests because, across the 5 aerosol sites, it is close to the 1991-1999 mean (mean value of 5 sites NRMSDs between yearly results and 1991-1999 mean is 45%, at Villum only it is 37%).
30 1992 is slightly closer to the 1991-1999 mean (37% NRMSD), but 1997 overlaps better with aerosol observations.

Is panel A in Figure 4 for 1997 only or for 1991-1999 (corresponding to Figure 2)? Based on this single year simulation, my understanding that authors choose the option with multiyear sea ice emissions decreased by 50% (panel C) for subsequent simulations (page 9, line 15). The authors should justify this. If this is the simulation they choose, it should be the one they show in Figure 3. To clarify the assumptions for the various simulations, the authors should include a table in the

5 supplementary material with the actual assumptions that are made. For example what salinity (over what sea ice) and snow age are used in Figure 4E?

The caption for Figure 4 has been altered. The text has been improved to describe the base simulation clearly, with reference to Table 1. The base simulation is introduced at the beginning of the Methods section (pg. 3, line 15).

10

7) Page 10 line 15. Do the authors have any potential explanations for why the observations at Barrow are reproduced by the SISS simulation during the first part of the year, but not the second part? Are the meteorological conditions (windspeed) not captured as well?

We speculate that the snow salinity on the sea ice surface should vary seasonally with the cycle of sea ice formation and melt (pg. 11, line 12). We don't know of any reason to doubt the ERA interim wind data at this location.

8) Seasonal variability of Na in ice cores. The authors compare the p-TOMCAT simulation to ice core observations over Greenland, finding that the model captures the observed seasonality with a winter maximum (section 4.3.2). Figure 5 shows20 that this seasonality is mostly due to the open ocean SS aerosol (dashed red line), with little influence from the sea ice SS sources. This is contrast to the open ocean (OOSS) simulation of atmospheric Na at ground sites in the Arctic (Figure 3). Can the authors explain the reason for this different modeled seasonality in the atmosphere and in ice cores for the OOSS simulation?

Most of the Arctic aerosol sites on Fig. 3 show little seasonality in OOSS but the model results for Summit do suggest seasonality in aerosol OOSS at that location. This is then in agreement with the model results for Greenland ice cores (including at Summit) on Fig. 5 (see pg. 14, lines 20-23). Because the Greenland ice cores are located further south than the aerosol sampling sites (excepting Summit, Fig. 1), they are influenced more strongly by OOSS sources. The reason for the simulated OOSS seasonality at Greenland ice core sites is difficult to isolate. It may be related to the seasonality of precipitation in the model, which controls wet deposition

30 occurrence. Please also see Sect 5, pg. 16, lines 6-14.

Also the comparison between p-TOMCAT and ice core measurements is a little difficult to follow as different sites are shown in different figures. For example, Tunu is missing from figure 5, but is shown in Figure 7. I suggest that the authors add Tunu in Figure 5, especially as it appears that the modeled influence of sea ice sources might be large at this site.

Tunu was not originally included on Fig. 5 because it is a relatively low accumulation site (~11 cm water/year), 5 meaning that the seasonal signal in [Na] is not as well-defined as at other sites, particularly in deeper (older) sections of the core. However, for the 1990s, the data look good and Tunu is the most northerly ice core with a relatively high proportion of SISS so the D5 ice core has been replaced with Tunu on Fig. 5.

9) Section 5. Based on the comparison shown in Figure 5, it seems that the sea ice sources do not really lead to a better10 simulation of the ice core measurements. At most sites the influence of sea ice sources is small. The largest modeled sea ice influence is at the NEEM site, where the model does not capture the observed seasonal cycle. Thus this comparison is inconclusive in terms of the role of a sea ice source in influencing ice core measurements.

The first sentence of section 5 has been re-phrased to emphasize that the importance of SISS to the ice core [Na] budget differs geographically, SISS does not make an "important contribution" in southern Greenland (pg. 15,

15 line 22). At NEEM the model captures the seasonality (max in winter, min in summer) in [Na], again the ice core [Na] peak might occur in early spring but be fixed to Jan 1st by the ice core dating technique. We believe the other statements in section 5 are sound and do not overstate the influence of SISS on Greenland ice core [Na].

Technical corrections:

20 - Page 8 line 20-22. This sentence is confusing. The Weddell sea salinity (0.3 psu) multiplied by two is 0.6 psu, while this sentence implies it is 0.12 psu. The Mundy observations of 0.1 psu of surface snow over the central Canadian Arctic thus imply that the salinity used by this study (0.6 psu?) is too large.

We agree, this is confusing. The *median* Weddell sea salinity x2 is 0.12 psu, which is similar to the *mean* salinity reported by Mundy for surface snow (0.11 psu). Please see Sect. 3.3.1 for a clearer description of observed and modelled snow salinity.

- Page 15 line 16. "SISS contributes to the winter maxima observed in all the ice cores, but that in some cases, OOSS alone can produce winter maxima and summer minima in sea salt in ice cores" There is no evidence of this in the manuscript. Figure 5 shows that OOSS reproduces the observed seasonal cycle at all sites except for NEEM. At NEEM, adding the SISS

30 source doesn't lead to a better simulation.

In the text the quotation above starts with the qualifying statement "Our simulations...suggest that...". Fig. 5 shows simulations that indicate SISS contributes to the winter [Na] maxima at ice core sites. Because adding the

additional SISS input in winter months causes the simulated values to increase further beyond the measured ice core values doesn't mean that the SISS input does not occur or is not important. At most sites the summer OOSS Na concentrations are higher than the ice core measurements, suggesting that deposition of OOSS over Greenland is over-estimated by p-TOMCAT. Please see Sections 4.3.2 and 5.

6

5

Other technical corrections have been addressed.

**Reviewer 2**

Rhodes et al. use a chemical transport model to examine the importance of the sea ice source of sea salt aerosol (SISS) relative to the ocean source of sea salt aerosol (OOSS) in the Arctic. They compare their model to observations of sea salt aerosol in the atmosphere and high resolution Na+ measurements in

- 5 Greenland ice cores. I found this paper very hard to follow and in the end it wasn't clear what was learned from their modeling exercise beyond what others have published.
  As is stated several times in the manuscript this is the first study to utilise a chemical transport model to simulate the sea salt concentrations of snow, therefore enabling direct comparison with ice core records (e.g., in abstract
- pg. 1 lines 14-15). Furthermore, we show that Na concentrations can be simulated to within a factor of 2. It will 10 be followed by an investigation into the factors driving signal variability (pg. 17, line 6) and studies using
  - paleoclimate boundary conditions (pg. 16, line 5). We trust that our manuscript is easier to follow now that the suggestions provided by reviewers are implemented.
- Because the processes responsible for the emission of SISS into the atmosphere are not well understood, the authors "tune" their model to best match the aerosol observations. In the discussion of all of the different parameters that can be tuned, the manuscript would greatly benefit first from an explicit description of the parameterization for SISS (i.e., show the actual equations, and define all of the variables). Without it, it is very hard to follow the discussion of the model tuning. We choose not to repeat the equations for SISS emission of Yang et al. (2008) because we do not introduce any

20 back to Yang et al. (2008).

It seems however that some of the model tuning has to do with the treatment of aerosol deposition in the model, not just the SISS emission parameterization, the discussion of which is also confusing. The parameterisation of sea salt deposition has not altered since Levine et al.'s (2014) work. What we describe in

25 section 2.3.3 is how the amount of sea salt *deposited* at each time step is calculated. Previously, only the amount *remaining in the atmosphere* after deposition was calculated. We do this in order to make a direct comparison with ice core sea salt concentrations (rather than atmospheric aerosol concentrations) (pg. 6, line 13). Please see our reply to Reviewer 1 for suggested changes that will make the SISS emission parameterisation easier to follow.

Not all of the terms in Equation 1 are defined. What is  $\alpha C$  PCL and  $\alpha N$  PNL? Is this somehow related to  $\alpha C$ ,  $\alpha N$ , PNL and PCL? It looks like there must be a mix up of subscripts and superscripts in either the equation or the text.

Yes, this is a typo mixing up subscripts and superscripts that has been rectified.

Does the model calculation of dry deposition include gravitational settling of the larger ( $r > 4 \mu m$ ) particles? If not, it should. Yes, see Pg. 6 line 12.

40 The modeled wet deposition seems to be missing some important processes (Page 13). It's also not clear if the modified snow precipitation directly influences wet deposition, or of the modeled wet deposition uses the "incorrect" precipitation.

The wet deposition scheme does have limitations, as we acknowledge in the text. The wet deposition code uses the model-generated precipitation (black line on Fig. 6 B-D) and this is now stated clearly in text (pg. 4, line 24).

I think what is new about this manuscript is the comparison of the model with Greenland ice core Na+

- 5 observations. However, this is probably the most ambiguous part of the paper, and it's not clear to me what they learned from this exercise. They are comparing modeled versus observed seasonality, although it seems that the seasonality of ice core Na+ is unclear as it was determined assuming constant snow accumulation rates, which is probably not consistent with reality. Also perhaps the seasonality is not well preserved in the observational record because of factors such as snow redistribution (page 14).
- 10 The seasonality in ice core Na is significant in all the Greenland ice cores shown on Fig. 5 (see green lines and uncertainty bars that denote interannual variability). This seasonality is preserved even though processes such as snow redistribution have likely impacted the ice core records.

What may be uncertain is the *monthly timing* of peak [Na] because when an ice core is dated by counting of annual layers in chemistry records [Na] is often assumed to peak Jan 1st. Or more accurately, the ratio of non-sea salt sulphur (nssS) (mostly from sulphate) to Na is used and the minimum is dated as Jan 1st. The timing of the

15 salt sulphur (nssS) (mostly from sulphate) to Na is used and the minimum is dated as Jan 1-\*. The timing of the [Na] may therefore be artificially fixed as Jan 1st, when it could vary by few months either side. Support for the winter timing of peak [Na] comes from sea salt measurements of Arctic aerosol (Fig. 3) and fresh snow e.g., at Summit (Fig. S5). Figure S5 now includes nssS:Na measured in snow at Summit. Please see pg. 13, lines 11-16.

20 In the end it seems that the model shows little skill at simulating the observed seasonality of ice core Na+... The model simulates summer minima in [Na] and maxima in either winter or spring. This is similar to the observed ice core [Na] seasonality, especially when we bear in mind that the Na peaks may have been used as winter (Jan 1st) markers in ice core dating, thereby artificially fixing the timing of maximum [Na]. The model simulates the absolute amplitudes of the seasonal cycle reasonably well (Table 3).

The second paragraph of the summary (section 6) I think attempts to articulate what they learned from the model/ice-core observation comparison, but I still cannot figure out what was learned from this exercise. Given that this is the main new contribution of this paper, the paper should be substantially revised to better articulate their scientific contribution. Section 6 has been rewritten.

30 Page 2 line 30: The last sentence of this paragraph needs a reference.

A publication discussing these results in detail is currently being prepared. Once published, the data set will be available online at the NERC Polar Data Centre. A DOI is being generated (expected by end June) where metadata will be made available now. The DOI will be included in the revised paper (pg. 5, line 9) so researchers can access the data easily in the future.

35 Page 5 Lines 16-17: Provide a justification for the choice of 0.3 psu.

0.3 psu is the mean of the salinity distribution of samples from the top 10 cm of the snow pack on sea ice in the Weddell Sea.

Page 5 Line 21 and elsewhere: What does "snow age" mean? This should be defined. It's not clear how this should impact SISS.

- 40 Please also see reply to Reviewer #1 and text of pg. 5, lines 18-24.
   Page 9 lines 8-9: How was scenario #3 parameterized? Did you simply reduce salinity by 50%?
   In scenario #3 the area of multi-year sea ice in each grid cell used in SISS emission calculation is halved. This has the effect of halving the SISS emissions from the multi-year ice in that grid cell. It is not precisely the same as reducing the snow salinity by 50%. Sea ice area is only halved for this calculation, it does not result in
- 45 additional area of open ocean. The text has been altered to clarify this (pg. 9, line 14). Page 9 Line 17: Define NRMSD the first time used.

Done. Page 11 line 1: Unfinished statement. What are you comparing the model simulations to? This sentence has been re-worded. Specify "snow accumulation" instead of just "accumulation" throughout the manuscript. 5 Done. Page 14 line 24: What is a "Greenland ice core simulation"? Do you mean model simulation? Done. Page 15 line 25-26: Be sure to specify that this is for today's climate. Perhaps it would be different in a different climate 10 The sentence has been changed from "in the Late Holocene" to "under present day conditions". Figure S1 should be in the main text We disagree. Fig. S1 presents comparisons of aerosol Na measurements and simulations at low latitudes and is not integral to the manuscript or its conclusions. When Figure 3 is presented in the text, it is not yet clear what your "base case" simulation is, which I 15 think is what the blue line is in the figure. This information should be presented in order. Yes, we agree, this is confusing and has been changed. Please see reply to Reviewer 1 comments. Figure 7: What are the yellow and other 3 green colors? The acronyms should be restated in the figure caption. The caption of Figure 7 has been changed. The other colours are other ice core records located within the same grid square in p-TOMCAT. 20 Figure 8: The model-observation comparison appears good here probably because of the large (2 order ofmagnitude) range in the color bar. The observations themselves cover a much smaller range, so the color bar should be scaled according to the range of the observations. Also I'm not sure this is the appropriate figure type to show because of the uncertainties in the SISS parameterizations. It would be best to have a figure that communicates the full

25 model range using all of your sensitivity simulations. Both scales are log scales to incorporate the wide range of values. Using a linear scale provided no useful information to the reader. The log scale means the variability at low values (most of the observations) is well represented in the colour scale. Readers can also consult Table 3 for the [Na] values of both model and ice core data.

30 We understand the reviewer's concerns about the sensitivity of our results to the SISS (and OOSS) parameterisations. We designed two experiments intended to produce extreme SISS:OOSS in order to provide a range of possible SISS:OOSS ratios. We have chosen not to include these data in the study because they use parameter choices that our sensitivity tests have already demonstrated produce unrealistic results i.e., the match between Arctic aerosol data and simulated values deteriorates. We have endeavored to be open about our choices

35 of parameters during tuning and demonstrated the effect of changing each of them though sensitivity tests. Ultimately, we have chosen a combination that produces the best match with the aerosol observations while acknowledging that other options may lead to similar results (pg. 10, line 27).

**Sea ice as a source of sea salt aerosol to Greenland ice cores: a model-based study**

Rachael H. Rhodes1, Xin Yang2, Eric W. Wolff1, Joseph R. McConnell3, Markus M. Frey2

1Department of Earth Sciences, University of Cambridge, Cambridge, CB2 3EQ, UK 2British Antarctic Survey, Natural Environment Research Council, Cambridge, CB3 0ET, UK

3Division of Hydrologic Sciences, Desert Research Institute, Reno NV, 89512, USA

Correspondence to: Rachael H. Rhodes (rhr34@cam.ac.uk)

Abstract. Growing evidence suggests that the sea ice surface is an important source of sea salt aerosol and this has significant implications for polar climate and atmospheric chemistry. It also suggests the potential to use ice 10 core sea salt records as proxies for past sea ice extent.\_To explore this possibility in the Arctic region, we use a chemical transport model to track the emission, transport and deposition of sea salt from both the open ocean and the sea ice, allowing us to assess the relative importance of each. Our results confirm the importance of sea ice sea salt (SISS) to the winter Arctic aerosol burden.\_For the first time, we explicitly simulate the sea salt 15 concentrations of Greenland snow, achieving values within a factor of two of Greenland ice core records. Our simulations suggest that SISS contributes to the winter maxima in sea salt characteristic of ice cores across Greenland. However, a north-south gradient in the contribution of SISS relative to open ocean sea salt (OOSS) exists across Greenland, with 50% of winter sea salt being SISS at northern sites such as NEEM (77°N), while only 10% of winter sea salt is SISS at southern locations such as ACT10C (66°N). Our model shows some skill at reproducing the inter-annual variability in sea salt concentrations for 1991-1999 AD, particularly at Summit 20 where up to 62% of the variability is explained. Future work will involve constraining what is driving this interannual variability and operating the model under different paleoclimatic conditions.

**1** Introduction**

25

5

Salty blowing snow lofted from the surface of sea ice may be an important source of sea salt aerosol to the polar atmosphere (Yang et al., 2008), with significant implications for climate and atmospheric chemistry.\_Sea salt aerosol act as cloud-condensation nuclei (O'Dowd et al., 1997) and ice nucleating particles (DeMott et al., 2016), impacting radiative forcing (Murphy et al., 1998), as well as providing surfaces for heterogeneous chemical reactions that impact the levels of key atmospheric trace gases, such as ozone (Knipping and Dabdub, 2003; Yang et al., 2010). For paleoclimatogists, this new source of sea salt provides a mechanism that links the sea salt

[revised manuscript text omitted]

Moved (insertion) [1] Deleted: For all of the ice cores,

**Deleted: 1**

| Deleted: (F | ïg. | 2 |
|-------------|-----|---|
|-------------|-----|---|

**Formatted: Subscript**

**2.3.2 Sea salt emissions**

Parameterisation of OOSS emissions follows Gong et al. (2003) and is based on the classic Monahan (1986) model of aerosol production via bubble bursting (Fig. 2).\_Gong's scheme is modified to account for a dependence of sea salt aerosol production on sea surface temperature (SST) (Eq. (4), Jaeglé et al. (2011)).

- 5 Parameterisation of SISS emissions follows Yang at al. (2008) (Eq. (1-8)) and this requires salinity and particle size distributions of snow particles entrained from the sea ice surface during blowing snow events to be defined (Fig. 2). We use new observations made during a winter-time cruise of the RV Polarstern (June-August 2013) in the Weddell Sea, Antarctica. These measurements were conducted in the framework of the BLOWSEA project led by the British Antarctic Survey (INCLUDE DOI HERE). The salinity distribution only includes
- 10 measurements from the top 10 cm of the snow pack, as this snow is the most likely to be lofted up. Any individual salinity measurements > 10 psu are excluded from the distribution. The mean salinity is 0.30 psu, which is 14-fold lower than that of the salinity distribution used by Levine et al. (2014) (4.25 psu) for snow on Antarctic sea ice. In our base simulation, this salinity distribution is doubled for snow on Arctic sea ice (Table 1, Sect. 3.3.1). The probability density function that defines the size distribution of suspended particles in blowing.
- 15 snow events (Yang at al.'s Eq. (6)) has a snow particle radius of 70.3 μm and shape parameter (α) value of 2. p-TOMCAT does not simulate snow particles splitting into multiple individual sea salt aerosol (cf. Huang and Jaeglé, 2017).

Yang at al.'s (2008) parameterisation of SISS production includes a parameter called snow agev(t in Yang at al.'s Eq. (5)), adopted from Box et al. (2004). A higher value of snow age decreases SISS emissions, loosely

- 20 representing how sintered snow flakes are likely more difficult to mobilise than fresh ones. Levine et al. (2014) found that the precipitation frequency and intensity within p-TOMCAT was not suitable for defining a transient snow age so a constant value of 5 days was used. When combined with our reduced snow salinity, this high snow age, which reduces the amount of blowing snow by almost a factor of 4 compared to a snow age of zero, resulted in extremely low SISS emissions. Since it is not clear that the parameterisation of snow age has any firm basis for
- 25 the very cold conditions encountered in the Arctic, we used snow age as a crude tuning device, and (as discussed in Sect. 3.3.3) adopted a value of 24 hr for our base simulation (Table 1). Finally, the 'gustiness factor' used by Levine et al. (2014) to increase the 6-hourly wind speeds used for sea salt aerosol emissions has been removed because it is specific to a different chemical transport model (Gong et al., 2003). We haven't replaced this value so peak sea salt emissions may be underestimated due to the 6-hourly

| {        | Deleted: We use                                                                   |
|----------|-----------------------------------------------------------------------------------|
| ····{    | Deleted: new                                                                      |
|          | Deleted: for the blowing snow particles entrained from the sea ice surface |
| $\geq$   | Deleted: . Both reflect                                                           |
| } | Deleted: ,                                                                        |
| {        | Formatted: Highlight                                                              |

| Deleted: | two-parameter gamma         |
|----------|-----------------------------|
| Deleted: | (Budd, 1966; Schmidt, 1982) |
| Deleted: | , which                     |
| Deleted: | Eq. (6),                    |
| Deleted: | (2008)                      |
| Deleted: | ,                           |
| Deleted: | parameter                   |

| Deleted: used a snow age |  |  |
|--------------------------|--|--|
| Deleted: ,               |  |  |
| Deleted: but w           |  |  |

|
|-----------------------------------|
|
|
|

averaging of wind speeds. Sensitivity testing indicates that using a 'gustiness factor' decreases the correspondence between model results and aerosol data at Arctic sites (Fig. S4).

**2.3.3 Sea salt deposition**

The deposition of OOSS and SISS in p-TOMCAT follows, the parameterisations of Reader and MacFarlane

- 5 (2003) (see also Levine et al. (2014) Eq. (1-9)). Wet deposition via nucleation and collision are both parameterised by exponential decay, Collision scavenging is determined by the collision scavenging parameter ( $\alpha_{C}$ , units:  $m^2 \text{ kg}^{-1}$ ) that varies with  $r_{wet}$  and by the rate of precipitation occurring at the same atmospheric level and all levels above (PCL, units: kg m-2 s-1). Nucleation scavenging is dependent on the nucleation scavenging parameter ( $\alpha_N$ , units of  $m^2 \text{ kg}^{-1}$ ) and the rate of precipitation occurring only within the same atmospheric level
- 10 (PNL, units: kg m-2s-1).\_Dry deposition only occurs in the surface layer of the model, which has a half-height (h, units: m) that varies between 23 and 36 m, depending on the geographic location and season.\_Calculation of the dry deposition velocity (vd, units: m s-1) accounts for the processes of sedimentation and turbulence.
   In order to compare our model simulations of Arctic sea salt aerosol to Greenland ice core Na concentrations, wd, calculate how much OOSS and SISS is deposited at each time step, in addition to keeping track of the mass
- 15 remaining in the atmosphere (M, units: kg).\_The mass of sea salt in each particle size bin ( $r_{dry}$ ) removed from each sigma-pressure level (L) in the atmosphere at each time step ( $\Delta t = 1800$  s) via wet (MWD, units: kg) and dry deposition (MDD, units: kg) is calculated by Eq. (1) and (2) respectively.

|    | $MWD_{L,rdry,t} = M_{L,rdry,t-\Delta t} \times e^{-(\alpha_C P C_L + \alpha_N P N_L) \Delta t}$ | (1) |  |
|----|-------------------------------------------------------------------------------------------------|-----|--|
| 20 | $MDD_{rdry,t} = M_{rdry,t - \Delta t} \times v_d \times \Delta t \ / \ h$                       | (2) |  |
|    | $SS_{mass, rdry,t} = MWD_{L, rdry,t} + MDD_{rdry,t}$                                            | (3) |  |
|    | $Na_{mass, rdry,t} = SS_{mass, rdry,t} \times 0.3906$                                           | (4) |  |
|    | $Na_{flux} = (Na_{mass} \times 1e^9) / a \times 12$                                             | (5) |  |
| 25 | $[Na]_{snow} = Na_{flux}/A$                                                                     | (6) |  |

After converting the mass of deposited sea salt  $(SS_{mass}, Eq. (3))$  to mass of Na (Eq. (4)), the flux of Na (Naflux, units:  $\mu g m^{-2} yr^{-1}$ ) from the atmosphere to the ice sheet is calculated via Eq. (5), where a = area of grid box (units:  $m^2$ ) and Namass is a monthly total Na mass deposited (units: kg). Naflux is then divided by the snow accumulation

[revised manuscript text omitted]

under the three different multi-year sea ice options is relatively small at Summit (Fig. S4) and therefore this choice does not greatly impact the sea salt budget of the atmosphere above Greenland.

**3.3.3 Snow age**

Higher values of snow age result in reduced SISS emissions. We tested the impact of decreasing the snow age in

- 5 the Arctic from 24 hr in our base simulation (Fig. 4A) to 12 hr (Fig. 4E) or to zero (Fig. 4F) for 1997 AD, For some sites, such as Barrow and Alert, ΔNa was reduced with a snow age of 12 hr or (Fig. 4E and Fig. 4F compared to Fig. 4A). The model-observations match across all the Arctic sites was reduced for both the J2hr and zero snow age (NRMSD increased). If we exclude Zeppelin from the calculation for 12 hr snow age, the NRMSD is similar that achieved for the base simulation using a snow age of 24hr. The maximum change in
- 10 monthly [Na] caused by setting the snow age to zero is a 70% increase in [Na] at Barrow in January (Fig. S5),

**3.4 Comparison between p-TOMCAT and GEOS-Chem**

The performance of p-TOMCAT can be further evaluated by comparing the simulation Arctic sea salt aerosol budget to that reported by Huang and Jaeglé (2017), who use the GEOS-Chem model (Table 3). In order to make

15 a direct comparison with their reported values, Table 3 reports values for 2005 AD only, which refer to sea salt aerosol, not just Na, and are for the Arctic region only (note: Lifetime in the Arctic region ≠ Lifetime in atmosphere).

For OOSS, the two models are broadly similar, with a tendency towards a higher burden, surface concentration, and Arctic lifetime in GEOS-Chem. For SISS, the emission rates are different between p-TOMCAT and GEOS-

- 20 Chem. p-TOMCAT emits ~ 5x more SISS in the 0.57  $\mu$ m <  $r_{dry}$  <= 4.5  $\mu$ m range than GEOS-Chem, while GEOS-Chem emits more than double the SISS of p-TOMCAT in the smaller particle size range. This difference is due to the tuning introduced by Huang and Jaeglé (2017) that causes each snow particle to produce 5 sea salt aerosol (whereas in p-TOMCAT, one snow particle equals one aerosol). The result is that deposition rates for large particles in p-TOMCAT are proportionally greater, while the burden and surface concentration are quite similar
- 25 between the two models. However, for the smaller particles, the surface concentration and burden of sea salt are significantly lower in p-TOMCAT, leading to an Arctic lifetime of 1.7 days versus 6.6 days in GEOS-Chem. J.ack of observations of snow on sea ice in the Arctic, and of sea salt aerosol produced during blowing events, makes it difficult to constrain many of the key parameters related to the blowing snow SISS emission process. Although we use a snow salinity distribution double that of Antarctic observations, a snow age of 24 hr and a

|  | $\mathbf{\Omega}$ |
|--|-------------------|
|  | ()                |
|  | •                 |

| Delet | red: in the Arctic                                         |
|-------|------------------------------------------------------------|
| Delet | ed: Villum                                                 |
| Delet | red: the lower                                             |
| Delet | ed: C                                                      |
| Delet | ed:                                                        |
| Delet | red: lower                                                 |
| Delet | red: , but                                                 |
| Delet | ed: i                                                      |
| Delet | ed:                                                        |
| Delet | ed: halving                                                |
| Delet | red: 25                                                    |
| Delet | red: 4                                                     |
| Delet | ed: .                                                      |
| Move  | d down [2]: Lack of observations of snow on sea ice in the |

Arctic, and of sea salt aerosol produced during blow on suite in the in the international of the sea salt aerosol produced during blowing events, makes it difficult to constrain many of the key parameters related to the blowing snow SISS emission process. Although we have chosen a snow salinity distribution double that of Antarctic observations, a snow age of 24 hr and a 50% reduction in SISS emissions from multi-year sea ice relative to first-year sea ice, we understand that a different combination of these parameters could effectively produce the same results.

Moved (insertion) [2]

50% reduction in SISS emissions from multi-year sea ice relative to first-year sea ice in our base simulation, we understand that a different combination of these parameters could effectively produce the same results,

**3.5 Importance of sea-ice-sourced sea salt aerosol**

Despite the somewhat ambiguous choices of parameters that we have to make, it is important to note that in all the individual sensitivity tests conducted for 1997 AD, SISS contributes to offset the winter OOSS Na deficit at all five Arctic aerosol sites (Fig. S5). For the full base simulation, the addition of SISS produces seasonal cycles that match well with overlapping Arctic aerosol observations. NRMSDs of between 34% for Villum and 89% for Alert (Fig. 3) are achieved. At Zeppelin on Svalbard, the modelled OOSS contribution is too high throughout the

year. However, the seasonal profile of SISS looks promising js amplitude is similar to the seasonal cycle of the
 observations. Villum, N. Greenland, shows the best model-observations agreement, with SISS contributing 80% of the total Na in the winter months on average. Results for Barrow, Alaska, are equally encouraging for January to June, but p-TOMCAT appears to underestimate SISS in the latter half of the year, hinting that SISS emission rates may vary with the cycle of sea ice decay and regrowth.

Only Alert, Canada, shows a significant offset between the aerosol observations and the modelled Na 15 concentration (Fig. 3). The summer concentrations, dominated by OOSS match well, but in other months p-TOMCAT underestimates [Na]. Huang and Jaeglé (2017) had a similar problem estimating aerosol [Na] at Alert

- and suggested that it results from Alert being situated in a region of relatively calm and stable meteorological conditions where the threshold wind speed ( $\sim$ 7 m s-1) for SISS emissions is not reached as often.\_Huang and Jaeglé (2017) found that the inclusion of an explicitly parameterised frost flower source (Xu et al., 2013) helped
- 20 to match the observed sea salt aerosol budget at Alert. Further field measurements are required to assess to what extent frost flowers do actually contribute aerosol to the atmospheric sea salt budget at low wind speeds, given evidence to the contrary (Obbard et al., 2009; Roscoe et al., 2011; Yang et al., 2017).

The simulated seasonal Na aerosol cycle for Summit, Greenland, matches the aerosol observations well (Fig. 3). Our results suggest that OOSS is the dominant source of Na to the high altitude central interior of the Greenland

25 ice sheet with significant SISS Na only present from November to March, contributing a maximum of 44% of the monthly Na budget.

| Deleted:                              |
|---------------------------------------|
|                                       |
| Deleted: 4                            |
|                                       |
| Deleted: se                           |
| Deleted: tuning                       |
| Deleted: five                         |
| Deleted: 4                            |
| Deleted:                              |
| Deleted: 1991–1999 AD                 |
| Deleted: s using the tuned parameters |
| Deleted: the                          |
| Deleted: well                         |
| Deleted: w                            |
| Deleted: ith                          |
| Formatted: Not Highlight              |
| Deleted: 90                           |
| Deleted: .                            |
| Deleted:                              |
| Deleted: I                            |
| Formatted: Not Highlight              |
| Deleted: 3                            |
| Deleted:                              |
| Deleted: (2016)                       |
| Deleted: 6                            |
| Deleted: ,                            |
| Deleted: but f                        |
| Deleted: work                         |

**4 Comparison of p-TOMCAT simulations to ice core Na records**

We now examine our p-TOMCAT base simulation (Table 1) of deposited sea salt for 1991–1999 AD to investigate the contribution of SISS to sea salt concentrations of Greenland ice core records. All the ice cores we consider are located at > 2000 m elevation and > 100 km inland (Table 2) so maximum Na concentrations are < 100 ppb. Seasonal variability in [Na] is consistently characterised by winter maxima and summer minima (Fig.

5); the amplitude of the mean seasonal cycle in the different ice cores varies between 6 and 55 ppb.

**4.1 Influence of snow accumulation rate**

5

Given that simulation of ice core Na concentrations using p-TOMCAT requires both the mass of Na deposited and the amount of precipitation at the ice core site (Eq. (6)), it is important that p-TOMCAT simulates

- 10 precipitation accurately, On the annual scale, the p-TOMCAT precipitation rates (forced towards GPCP observations, Sect. 2.3.1) agree well with ice core snow accumulation rates (Fig. 6A). Northern sites like NEEM and Tunu show model-ice core agreement to within 30%. Summit annual mean snow accumulation rate is estimated to within 2%. Further south, the model-ice core agreement reduces as p-TOMCAT has trouble capturing the steep gradient in snow accumulation rate between the coast and the interior of the ice sheet over
- 15 Southern Greenland.\_At ACT11d, for example, the simulated precipitation, rate is 250% higher than that suggested by the ice core.

The simulated precipitation rate at a single Greenland ice core site can vary by a factor of 4 across a year (Fig. 6B–D). At NEEM in northwest Greenland, the simulated precipitation rate is consistently higher in summer relative to winter (Fig. 6B), whereas at Summit in central Greenland the simulated precipitation rate is greater in

- 20 winter relative to summer (Fig. 6C). Ice core sites further south don't show a clear seasonal signal in modelcalculated precipitation rate (Fig. 6D). We have a small amount of information about how snow accumulation rates over Greenland vary seasonally. Recent field measurements at Summit (2003–2014 AD) agree with satellite-based laser altimetry measurements, indicating that the monthly accumulation rates are highly variable with a tendency towards relatively low snow accumulation in spring and relatively high snow accumulation in
- 25 autumn (Fig. 6C), Other work, focused on the Summit, NGRIP and NEEM sites, found evidence for a summerweighted bias in snow accumulation (Shuman et al., 1995, 2001; Steen-Larsen et al., 2011), suggesting p-TOMCAT may in fact be doing a good job of representing seasonal accumulation variability in northern Greenland (Fig. 6B).

|   | - |
|---|---|
| 1 | ~ |
|   |   |
|   | ~ |

**Deleted:**

|    | Deleted: over the polar regions |
|----|---------------------------------|
|    |                                 |
| -1 | Deleted: values                 |
|    |                                 |
|    |                                 |
|    |                                 |

|
|-------------------------------|
|                               |
|
|                               |
|

|
|---------------------------------------------------------------------|
|
|

We test the effect of substituting the constant monthly ice core snow accumulation rate for A in Eq. (6) when calculating the Na concentration of snow falling at the ice core sites because a constant rate of snow accumulation per year was assumed when dating the ice core records  $_{\rm m}$ This does not remove all possible bias due to the modelled precipitation seasonality because the modelled precipitation is still in wet deposition calculations

- 5 (Eq. (1)). Simulated ice core [Na] calculated by this method are displayed on Fig. 5B, and simulated ice core [Na] calculated using the model-calculated snow accumulation rate in Eq. (6) are displayed on Fig. 5A. At ice core sites where accumulation rates are over-estimated by p-TOMCAT, i.e., D4, D5, Das2 and S. Greenland (ACT10C, ACT3 and Das1), Na concentrations broadly increase when the [lower] ice core accumulation rate is used (Fig. 5B compared to Fig. 5A). Modelled precipitation for Summit (Fig. 6C), D4, D5
- 10 and Das2 is lower in April to June relative to other months causing a prominent spring-early summer maximum in simulated Na, specifically OOSS (Fig. 5A). Using the constant ice core accumulation rate this feature disappears and the [Na] maximum occurs in the winter months, in agreement with the ice core data seasonality (Fig. 5B), It is possible that the assumption of winter timing of [Na] peaks made in ice core dating is incorrect and that [Na] seasonality in Greenland ice cores is actually like the simulated profiles on Fig. 5A. However, this
- 15 seems unlikely because [Na] values of Greenland aerosol (Fig. 3) and fresh surface snow at Summit (Fig. SQ peak in the winter months.

**4.2 Smoothing of the snowpack Na signal**

Comparison between p-TOMCAT [Na] simulations and Greenland ice core records reveal significant month-tomonth variability in the simulated time series that is not present in the ice core records, which are all

- 20 characterised by smoothly oscillating [Na] with a clear seasonality (Fig. 7). We hypothesise that the deposited Na signal is smoothed by surface snow redistribution by winds and compaction of the snow pack during densification (Dibb and Jaffrezo, 1997). Evidence for this smoothing process comes from comparison of Na concentrations of weekly surface snow samples at Summit and ice core [Na] measurements dating from the same time interval (Fig. S6). The surface snow Na concentrations are much more variable with rapid, large (~20 ppb)
- 25 oscillations, However, the timing and magnitude of the underlying seasonal cycle corresponds well with the ice core record. The ice core [Na] signal may also be damped by dispersive mixing within the continuous analysis system (Breton et al., 2012), specifically for lower snow accumulation sites such as Tunu. We crudely represent the cumulative effect of these smoothing processes by applying a Savitzky-Golay filter (span = 4%, order = 2) to the simulated [Na] time series (Fig. 7). The stacked simulated [Na] seasonal cycles for 1991–1999 AD are 30 displayed on Fig. 5. Unfiltered Na seasonal cycle stacks are displayed on Fig. S6.

|   | - |  |
|---|---|--|
|   |   |  |
|   |   |  |
|   | • |  |
| L | ~ |  |
|   |   |  |

|    | Deleted: We should remember that a constant rate of accumulation per year was assumed when dating the ice core records. With this in mind, |
|----|---------------------------------------------------------------------------------------------------------------------------------------------------|
| Ì  | Deleted: w                                                                                                                                        |
| {  | Deleted:                                                                                                                                          |
| 1  | Deleted:                                                                                                                                          |
| Ù  | Deleted: account                                                                                                                                  |
| Ĵ  | Deleted: for                                                                                                                                      |
| Ŭ  | Deleted: in the model                                                                                                                             |
| Ň  | Deleted: of sea salt is controlled by precipitation occurrence and amount                                                                  |
| Ň  | Deleted:                                                                                                                                          |
| ľ  | Deleted:                                                                                                                                          |
| -1 | Deleted: reduced                                                                                                                                  |

| D | eleted: [Na]              |
|---|---------------------------|
| D | eleted: measurements of   |
| D | eleted: S5                |
| D | eleted: [Na] is           |
| D | eleted: in concentrations |
| D | eleted: , but             |
| D | eleted: overall           |

**4.3 How well are Greenland ice core records represented by p-TOMCAT?**

**4.3.1 Annual mean**

The majority of Greenland ice core annual mean [Na] values (1991–1999 AD) are simulated to within a factor of 2 by p-TOMCAT (Fig. 8A, Table 4). Tunu, NEEM and ACT10C annual means are simulated most accurately,

5 regardless of the accumulation rate used to calculate the simulated [Na] (Table 4 and Table S1). Das2 in southeast Greenland and ACT11d and ACT2 is southwest Greenland are the most poorly simulated with p-TOMCAT over-estimating the extremely low ice core annual mean [Na] values of 5–8 ppb by  $\geq$  350% (Fig. 8A, Table 4). p-TOMCAT severely over-estimates the accumulation rate for these sites (Fig. 6A), suggesting that too much sea salt is deposited by wet deposition.

**10 4.3.2 Seasonal cycle**

p-TOMCAT is also successful in simulating the amplitude of the seasonal cycle of [Na] in the majority of Greenland ice cores to within a factor of 2 (Fig. 8B, Table 4) giving us confidence that p-TOMCAT is simulating meaningful seasonal variability. Again, the northerly sites are simulated most accurately: Tunu to within 1 ppb and Summit to within 4 ppb (Table 4). The seasonal cycles in the southern cores of ACT2 and ACT11d (Table 4)
 are over-estimated, which can be linked to the high simulated snow accumulation rates.

- At central and southern sites simulated summer (JJA) [Na] values are higher than the ice core data (Table 4), often by a factor of 5 or more, but we note that summer ice core [Na] values can be as low as 1 ppb\_It is interesting that the summer OOSS contribution to the ice core budget is over-estimated by p-TOMCAT because simulated aerosol OOSS concentrations in the surface layer of the atmosphere at Villum, Barrow, and Alert
- 20 appear to match summer observations well (Fig. 3), At Summit, correspondence with summer observations is greatly improved if the full 1991–2006 AD simulation mean is considered (not shown). We suspect this difference between aerosol and ice core simulations results from the simplistic deposition scheme of p-TOMCAT, which allows super-micron sized OOSS particles to be transported to the ice sheet and wet-deposited from high levels in the atmosphere (Fig. S3), The deposition scheme does not differentiate between in-cloud and
- 25 below-cloud scavenging rates (Zhang et al., 2013) and wet deposition rates are the same when precipitation is snow or rain (Wang et al., 2014). There is also no explicit consideration of fog deposition, which is common on the Greenland ice sheet (Bergin et al., 1995).

[revised manuscript text omitted]